# Differential inputs to striatal cholinergic and parvalbumin interneurons imply functional distinctions

Jason R Klug, Max D Engelhardt, Cara N Cadman, Hao Li, Jared B Smith, Sarah Ayala, Elora W Williams, Hilary Hoffman, Xin Jin*

Molecular Neurobiology Laboratory, The Salk Institute for Biological Studies, La Jolla, United States

**Abstract** Striatal cholinergic (ChAT) and parvalbumin (PV) interneurons exert powerful influences on striatal function in health and disease, yet little is known about the organization of their inputs. Here using rabies tracing, electrophysiology and genetic tools, we compare the whole-brain inputs to these two types of striatal interneurons and dissect their functional connectivity in mice. ChAT interneurons receive a substantial cortical input from associative regions of cortex, such as the orbitofrontal cortex. Amongst subcortical inputs, a previously unknown inhibitory thalamic reticular nucleus input to striatal PV interneurons is identified. Additionally, the external segment of the globus pallidus targets striatal ChAT interneurons, which is sufficient to inhibit tonic ChAT interneuron firing. Finally, we describe a novel excitatory pathway from the pedunculopontine nucleus that innervates ChAT interneurons. These results establish the brain-wide direct inputs of two major types of striatal interneurons and allude to distinct roles in regulating striatal activity and controlling behavior.

DOI: https://doi.org/10.7554/eLife.35657.001

*For correspondence:
xjin@salk.edu

## Introduction

Successful behavior requires the proper control of actions based on the integration of a wide variety of information, including both environmental stimuli and internal state, being processed across numerous brain regions. The basal ganglia, particularly the striatum, its main input nucleus, is a major node in the integration of multiple cortical and subcortical inputs underlying action control and learning (*Graybiel, 2000*; *Hikosaka et al., 1999*; *Jin and Costa, 2015*; *Yin and Knowlton, 2006*). The striatum is primarily composed of spiny projection neurons (SPNs), which consist of ~95% of total striatal neuronal population (*Gerfen et al., 1990*). Yet, intermixed throughout the striatum are local interneurons, which exert powerful regulation on SPN activity (*Gittis et al., 2010*; *Goldberg et al., 2012*; *Schulz and Reynolds, 2013*; *Silberberg and Bolam, 2015*; *Tepper et al., 2010*). This interneuron population includes large, aspiny choline acetyltransferase (ChAT)-positive cholinergic interneurons as well as a multitude of GABAergic interneuron subtypes differentiated based on expression of parvalbumin (PV), somatostatin, calretinin and other neurochemical markers (*Tepper et al., 2010*). Numerous studies have suggested that striatal ChAT and PV interneurons exhibit different membrane properties, connectivity and effects on modulating SPN activity (*Gittis et al., 2010*; *Lim et al., 2014*; *Silberberg and Bolam, 2015*; *Tepper et al., 2010*). However, compared to the different subtypes of SPNs (*Smith et al., 2016*; *Wall et al., 2013*) or the interneurons in cerebral cortex or hippocampus (*Freund and Buzsáki, 1996*; *Kepecs and Fishell, 2014*; *Wall et al., 2016*), relatively little is known about the organization of inputs to striatal interneurons let alone their precise function.

While ChAT interneurons account for only 1–2% of the total striatal population, the striatum contains some of the highest levels of cholinergic markers in the brain. Dysfunction of striatal ChAT interneurons has been implicated in numerous psychiatric disorders including schizophrenia, depression, and other mood disorders (*Scarr et al., 2013*), yet we lack a complete understanding regarding the input architecture to ChAT interneurons and their role in modifying behavior. ChAT interneurons are tonically active and exhibit a pause and subsequent rebound firing in response to the presentation of a cue predictive of reward or aversion (*Ding et al., 2010*; *English et al., 2012*; *Kimura et al., 1984*; *Schulz and Reynolds, 2013*). This conditioned pause is thought to encode the salience of external stimuli supporting the association of cue with action or outcome (*Aosaki et al., 1994*; *Kimura et al., 1984*; *Ravel et al., 1999*). Recently, striatal ChAT interneurons have been linked with behavioral flexibility and monitoring environmental state (*Aoki et al., 2015*; *Apicella, 2007*; *Bradfield et al., 2013*; *Brown et al., 2010*; *Okada et al., 2014*; *Prado et al., 2017*; *Stalnaker et al., 2016*). A vast majority of excitatory inputs to ChAT interneurons are thought to originate from centromedian and parafascicular nuclei of the thalamus (*Ding et al., 2010*; *Lapper and Bolam, 1992*); however, cortical stimulation has been shown to increase ChAT interneuron firing as well (*Ding et al., 2010*), suggesting a wealth of underappreciated cortical inputs.

PV interneurons (also known as fast-spiking interneurons) constitute ~1% of the total striatal neuronal population and their dysfunction has been implicated in multiple movement disorders and neuropsychiatric disorders including Huntington's disease, dystonia, obsessive-compulsive disorder (OCD) and Tourette's syndrome (TS) (*Burguière et al., 2015*; *Kalanithi et al., 2005*). Recent studies have suggested that PV interneurons increased their activity during action sequences and sensory-based decision making; however, their exact function in controlling behavior remains to be clearly determined (*Adler et al., 2013*; *Gage et al., 2010*; *Jin et al., 2014*). These neurons are capable of firing at very high rates *in vitro* and *in vivo* (*Gage et al., 2010*; *Jin et al., 2014*) providing strong feedforward inhibition onto SPNs (*Gittis et al., 2010*; *Koós and Tepper, 1999*). Striatal PV interneurons are believed to receive inputs primarily from cortex and globus pallidus, with minimal thalamic innervation (*Bevan et al., 1998*; *Mallet et al., 2012*; *Saunders et al., 2016*). However, the inputs to striatal PV interneurons have not been exhaustively mapped, which might be crucial for understanding their function in controlling behavior.

Here using rabies-mediated monosynaptic retrograde tracing and electrophysiology with optogenetics, we compare and analyze the whole-brain direct inputs to striatal ChAT and PV interneurons. Overall, ChAT and PV interneurons receive a vast majority of their inputs from the cortex. ChAT interneurons were found to preferentially receive inputs from association areas of cortex and thalamus. Among a wealth of different input targets revealed in the tracing study, we focus on the functional validation of three novel or underappreciated inputs to ChAT and PV interneurons. Specifically, a previously unknown inhibitory input from the thalamic reticular nucleus targeting PV interneurons was identified and functionally characterized. Additionally, an inhibitory pathway from the external segment of the globus pallidus to ChAT interneurons is explored, which robustly generates a pause in ChAT interneuron activity. We also identify a direct excitatory input from the pedunculopontine nucleus that targets ChAT interneurons. These results provide cell type-specific anatomical and functional connectivity for two major types of striatal interneurons providing insight into their role in controlling behavior.

## Results

### Monosynaptic tracing reveals the inputs to striatal ChAT and PV interneurons

We used transgenic ChAT-Cre and PV-Cre mouse lines to target striatal ChAT and PV interneurons in the striatum. To validate Cre line specificity, a Cre-dependent AAV virus that expresses eGFP was injected into the dorsal striatum of either ChAT-Cre or PV-Cre mice (*Figure 1a*). The eGFP expression in striatum of ChAT-Cre and PV-Cre mice was highly specific as demonstrated by colocalization with immunohistochemical staining for choline acetyltransferase or parvalbumin (*Figure 1a,b*; *Figure 1—figure supplement 1a–c*, ChAT 95.9 ± 0.78%, PV 95.6 ± 0.79%). Additionally, *ex vivo* electrophysiological recordings of GFP-expressing ChAT or PV interneurons show typical electrophysiological properties (*Figure 1c*). ChAT interneurons are tonically active, have depolarized

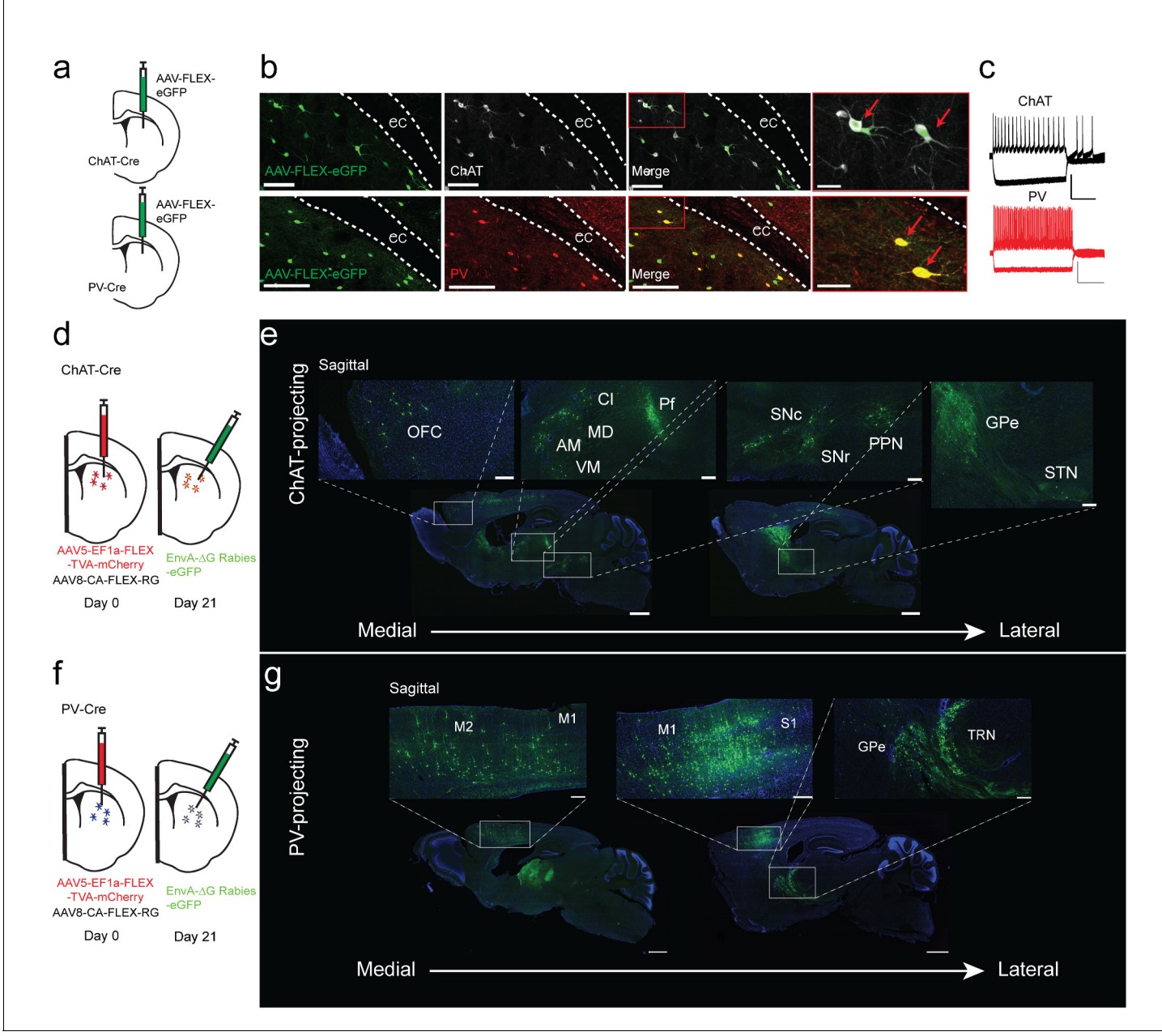

**Figure 1.** Monosynaptic tracing reveals the inputs to striatal ChAT and PV interneurons. (**a**) Schematic of Cre-dependent AAV-eGFP viral injection in ChAT-Cre or PV-Cre mice. (**b**) AAV-eGFP expression is highly colocalized with ChAT and PV immunostaining, respectively. Scale bars, 100 μm. inset (red box), scale bar 25 μm. Red arrows denote colocalization. ec, external capsule. (**c**) Representative traces of (top) ChAT and (bottom) PV interneuron to hyperpolarizing and depolarizing current injection (step −250 pA,+250 pA). Scale bars, 250 ms, 50 mV. (**d**) Schematic of Cre-dependent AAV helper viruses and modified rabies virus injections in ChAT-Cre mice. (**e**) Series of representative sagittal sections containing inputs to ChAT interneurons. (**f**) Schematic of Cre-dependent AAV helper viruses and modified rabies virus injections in PV-Cre mice. (**g**) Series of representative sagittal sections containing inputs to PV interneurons. Only the injection hemisphere is shown. Scale bars, 1 mm; inset scale bars; 500 μm. Brain regions are highlighted in white lettering. OFC, orbital frontal cortex; M1, primary motor cortex; M2, secondary motor cortex; S1, primary somatosensory cortex; GPe, globus pallidus external segment; Cl, central lateral thalamic nuclei; Pf, parafascicular thalamic nucleus; AM, anteromedial thalamic nucleus; MD, mediodorsal thalamic nucleus; VM, ventromedial thalamic nucleus; STN, subthalamic nucleus, SNc, substantia nigra pars compacta; SNr, substantia nigra pars reticulata; PPN, pedunculopontine nucleus; TRN, thalamic reticular nucleus. The following figure supplements are available for *Figure 1*: *Figure 1— figure supplement 1* and *Figure 1—figure supplement 2* for additional coronal images and starter cell quantification, respectively.
DOI: https://doi.org/10.7554/eLife.35657.002

The following figure supplements are available for figure 1:

**Figure supplement 1.** Overview of inputs to ChAT and PV interneurons.

*Figure 1 continued on next page*

*Figure 1 continued*

DOI: https://doi.org/10.7554/eLife.35657.003

**Figure supplement 2.** ChAT and PV starter cells are restricted to dorsal striatum, similar in total number and are distributed equally across dorsal medial and dorsal lateral striatal subdivisions.

DOI: https://doi.org/10.7554/eLife.35657.004

resting membrane potentials, and feature prominent hyperpolarization-activation cation currents (*Figure 1c*), while PV interneurons exhibit narrow action potentials and high firing rates (*Gittis et al., 2010*; *Tepper et al., 2010*). These results demonstrate that the ChAT-Cre and PV-Cre lines are highly selective in striatum and therefore appropriate for investigating the inputs to striatal ChAT and PV interneurons.

To determine the differences in inputs to striatal ChAT and PV interneurons, we performed Cre-dependent, modified rabies virus tracing (*Guo et al., 2015*; *Wickersham et al., 2007*) in the dorsal striatum of ChAT- or PV-Cre mice from unilateral viral injections into the same location (See Materials and methods, *Figure 1d–g*; *Figure 1—figure supplement 1d–i*; ChAT n = 6, PV n = 5). All starter cells were found to be restricted to the dorsal striatum without any cortical expression (*Figure 1—figure supplement 2a,b*). There was no difference in the total number or the striatal subregion distribution between ChAT and PV starter neurons (*Figure 1—figure supplement 2c–g*). In both Cre mice, eGFP-positive projection neurons were found throughout cortex, thalamus, basal ganglia and other subcortical regions (*Figure 1e,g*; *Figure 1—figure supplement 2f,g*). Labeling on the ipsilateral side throughout the brain was quantified relative to brain region boundaries (as defined by the Allen Institute Mouse Brain Reference Atlas). There was no significant difference in the total number of input neurons for ChAT and PV interneurons (ChAT 6580 ± 2657, PV 4456 ± 1540; two-tailed t-test, p=0.4773). A cutoff of greater than 0.4% was used to distinguish between major and minor inputs to these interneuron populations (*Figure 2a* for all major inputs, see a complete list in *Figure 2—figure supplement 1* with a statistical comparison between ChAT and PV inputs).

The tracing results revealed that both types of striatal interneurons received a vast majority of inputs from the cortex (*Figure 2b*, two-way ANOVA, $F_{(3, 36)}$=325.1, p<0.0001), with the remaining inputs coming from various thalamic nuclei, other basal ganglia nuclei, and a variety of other subcortical regions. ChAT interneurons received substantial inputs from cingulate cortex and secondary motor cortex. PV interneurons received extensive inputs from similar cortical regions with the addition of a substantial primary motor and primary somatosensory cortical inputs (*Figure 2a*). We observed input neurons to ChAT or PV interneurons from layer 2/3, 5, and 6 of cortex (*Figure 2c*). ChAT interneurons received significantly more inputs from layer five compared to all other cortical layers and more layer 2/3 compared to layer six projections (*Figure 2c*; two-way ANOVA, Sidak's, 5 vs. 6, p<0.0001; 2/3 vs. 6, p=0.0019; 2/3 vs. 5, p=0.0041).

When grouping the cortical inputs into association-like (e.g. orbital, insula, prelimbic, cingulate, M2, S2, RS, and PPC, see Materials and methods) and sensorimotor-like regions (e.g. M1, S1, AUD, VIS), ChAT interneurons received significantly more inputs from associative than sensorimotor cortex (*Figure 2d*, two-way ANOVA, Sidak's multiple comparisons test, p=0.0001). Examination of the laminar distribution in associative versus sensorimotor cortical regions revealed a unique pattern. Inputs in associative cortex are superficial preferring (layer 2/3) while sensorimotor inputs are more prominent in deeper layers (layer 5/6) (*Figure 2e,f*) (*Smith et al., 2016*). Association cortex inputs to ChAT interneurons were layer 2/3 dominant with significantly more layer 2/3 inputs than layer six and more layer five inputs than layer 6 (*Figure 2e*; two-way ANOVA, Sidak's, layer 2/3 vs. 6, p<0.0001, layer 5 vs. 6, p=0.0012). Similarly, PV interneurons association cortical inputs were more layer 2/3 preferring (*Figure 2e*; two-way ANOVA, Sidak's, layer 2/3 vs. 6, p<0.0001, layer 2/3 vs 5, p=0.0029). Sensorimotor inputs to ChAT interneurons originated predominately from layer 5 (*Figure 2f*; two-way ANOVA, Sidak's, layer 5 vs. 6, p=0.0010, layer 2/3 vs. 5, p=0.0086. PV interneurons also received significantly more sensorimotor layer 5 than layer 2/3 projections (*Figure 2f*; two-way ANOVA, Sidak's, layer 2/3 vs. 5, p=0.0018). These results show that although both types of striatal interneurons receive more superficial association cortex inputs and more deep layer sensorimotor cortex inputs, ChAT interneurons receive more associative cortical inputs overall.

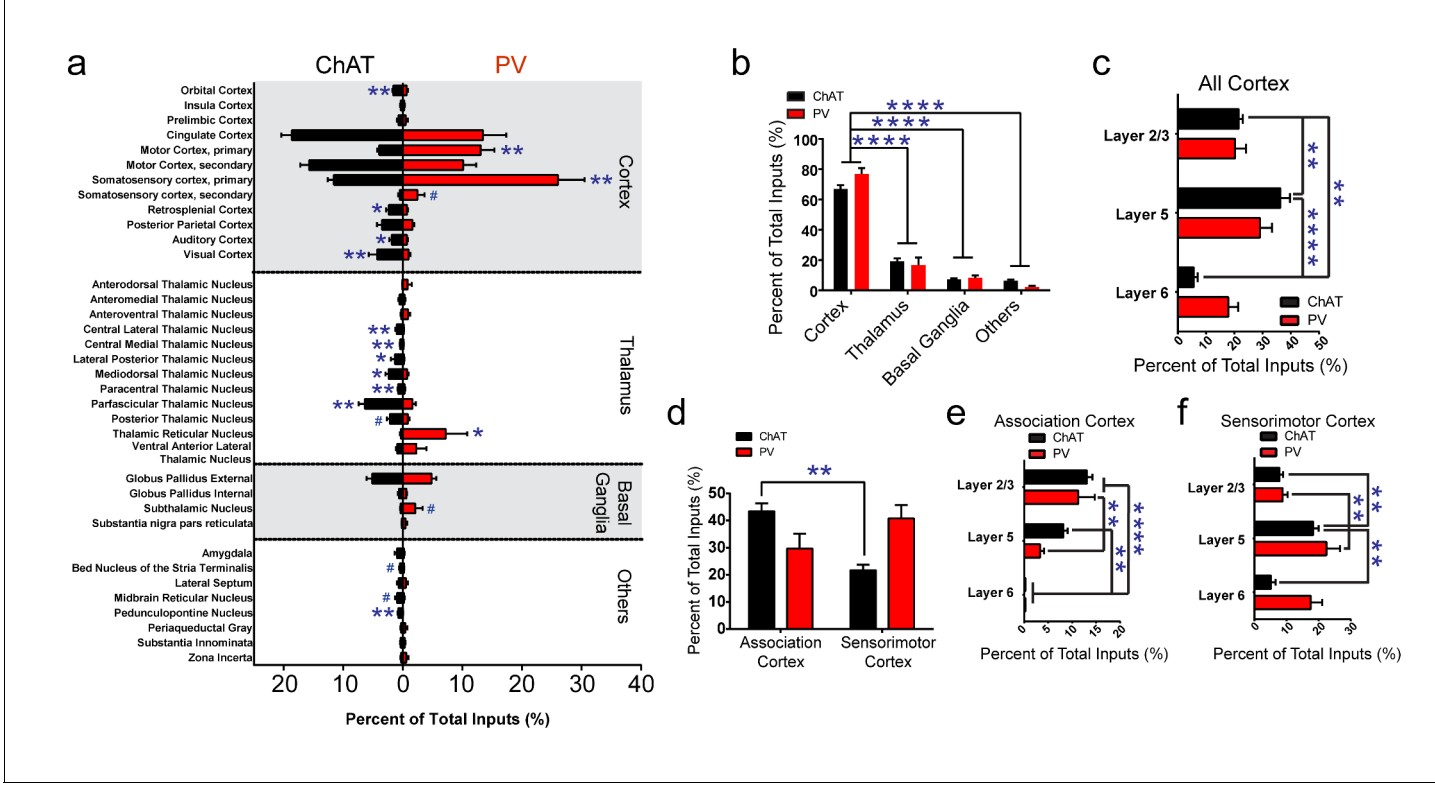

**Figure 2.** Summary of brain-wide direct inputs to striatal ChAT and PV interneurons. (**a**) Major input regions (>0.4% of total inputs) to ChAT (black bar) and PV (red bar) interneurons. A complete set of all inputs to ChAT and PV interneurons is shown in *Figure 2—figure supplement 1*. (**b**) ChAT and PV interneurons receive predominately cortical innervation (two-way ANOVA, $F_{(3,36)}$ = 325.1, p<0.0001). (**c**) Cortical laminar distribution of all inputs to ChAT and PV interneurons. ChAT interneurons receive a significant bulk of inputs from layer five compared to other cortical layers (two-way ANOVA, Sidak's, 5 vs. 6, p<0.0001; 2/3 vs. 6, p=0.0019; 2/3 vs. 5, p=0.0041). (**d**) Sorted association and sensorimotor cortex inputs to ChAT and PV interneurons (see Materials and methods). ChAT interneurons receive a greater percentage of associative versus sensorimotor inputs (two-way ANOVA, Sidak's, p=0.0001). (**e**) Association cortex laminar distribution is biased to superficial layers. ChAT interneurons receive more associative layer 2/3 than layer six projections and more layer 5 than six projections (two-way ANOVA, Sidak's, layer 2/3 vs. 6, p<0.0001, layer 5 vs. 6, p=0.0012). PV interneurons also receive more associative layer 2/3 than layer 5 and 6 projections (two-way ANOVA, Sidak's, layer 2/3 vs. 6, p<0.0001, layer 2/3 vs 5, p=0.0029). (**f**) Sensorimotor cortex laminar distribution is biased to deep layers. ChAT interneurons receive more sensorimotor layer five projections than layer 2/3 or layer six projections (two-way ANOVA, Sidak's, layer 5 vs. 6, p=0.0010, layer 2/3 vs. 5, p=0.0086). PV interneurons receive more sensorimotor layer 5 than layer 2/3 projections (two-way ANOVA, Sidak's, layer 2/3 vs. 5, p=0.0018). All figures mean ± SEM. In blue, *p≤0.05, **p≤0.01, ****p≤0.0001, # p<0.08. The following figure supplements are available for *Figure 2*: *Figure 2—figure supplements 1* and *2*.
DOI: https://doi.org/10.7554/eLife.35657.005

The following figure supplements are available for figure 2:

**Figure supplement 1.** Complete list of all inputs to ChAT and PV interneurons.
DOI: https://doi.org/10.7554/eLife.35657.006

**Figure supplement 2.** Summary of the inputs and proposed function of striatal ChAT and PV interneurons.
DOI: https://doi.org/10.7554/eLife.35657.007

## Thalamic projections to striatal ChAT and PV interneurons

The previous literature on thalamostriatal projections suggests that intralaminar nuclei mainly target striatal ChAT interneurons or SPNs (*Ding et al., 2010*; *Smith et al., 2004*), however a growing body of work also suggest intralaminar thalamic nuclei also target PV interneurons (*Arias-García et al., 2017*; *Assous et al., 2017*; *Sciamanna et al., 2015*; *Sidibé and Smith, 1999*). While all four intralaminar nuclei had inputs targeting ChAT interneurons, our dG-rabies tracing revealed that PV interneurons also received significant projections from intralaminar thalamus (*Figure 2a*). Interestingly, ChAT interneurons have inputs from all intralaminar nuclei and associative thalamic nuclei like mediodorsal thalamus (*Figure 2a*). Together, these data suggest that like association cortex, associative

thalamic nuclei and intralaminar nuclei also have a substantial proportion of inputs targeting striatal ChAT interneurons.

Our dG-rabies tracing surprisingly revealed that PV interneurons receive a projection from the thalamic reticular nucleus (TRN) (*Figure 2a*, *Figure 3a*). The TRN is a thin shell of GABAergic neurons that was thought to project within thalamus regulating thalamocortical and corticothalamic communications (*Halassa and Acsády, 2016*). The rabies-eGFP signal was clearly located within the TRN as evidenced by colocalization with parvalbumin (PV) or somatostatin (SOM) immunostaining (*Clemente-Perez et al., 2017*) (*Figure 3b*, *Figure 3—figure supplement 1e,f*, PV: rabies-eGFP colocalization 33.44 ± 6.08%, SOM: rabies-eGFP colocalization 43.73 ± 3.88%), and was enriched in the most rostral pole of the TRN. Noticeably, the labeling of TRN was never observed in control experiments with either helper or rabies virus alone nor were TVA-mCherry starter cells observed in TRN (*Figure 1—figure supplement 1h,i*). In order to anatomically validate this projection from TRN to PV interneurons independently, three additional viral tracing experiments were performed. In the first approach, anterograde tracing from TRN in PV-Cre mice with a Cre-dependent AAV expressing eGFP (*Figure 3c*), showed that labeled TRN neurons sent projections coursing rostrally into the striatum and form close appositions to PV interneuron somas and proximal dendrites (*Figure 3d*). Secondly, using a convergent intersectional approach to trace TRN projections, a retrograde AAV virus (*Tervo et al., 2016*) carrying Flp-recombinase was injected into dorsal striatum of a PV-Cre mouse, while a dual necessity Cre-/Flp- recombinant INTRSECT AAV virus was injected in TRN (*Figure 3e*). Following transfection, eYFP-expressing cell bodies were seen in the anterior TRN, but not surrounding areas like GPe, with clear axonal projections observed in the dorsal thalamus (*Figure 3f*). Utilizing striatal PV immunostaining, eYFP fibers from TRN were observed to be in close apposition to PV cell somas in the striatum (*Figure 3f*). As an additional confirmation of TRN to striatal connectivity we injected Somatostatin-IRES-Cre mice with Cre-dependent AAV-FLEX-eGFP to label TRN neurons (*Figure 3—figure supplement 1g–i*). TRN expressed eGFP selectively and eGFP-positive fibers can be seen near PV interneurons in striatum. Thus, these three viral tracing experiments have independently verified the rabies tracing results and confirmed a previously unknown projection from TRN to striatal PV interneurons.

To further test the functional connectivity of this projection from TRN to striatal PV interneurons, whole-cell voltage clamp recordings from PV interneurons following TRN terminal stimulation were conducted (*Figure 3g*). Brief paired blue laser stimulation of TRN ChR2-positive terminals in the striatum over PV interneurons, in the presence of AMPAR and NMDAR antagonists, exhibited fast latency, monosynaptic IPSCs in 53.8% (7/13) of the cells recorded (*Figure 3h–j*). These large, reliable, and paired-pulse depressing IPSCs were blocked by the GABA$_A$R antagonist picrotoxin (*Figure 3h*). Connectivity was not determined by location as PV interneurons in close proximity to other connected PV interneurons did not show connectivity in the same brain slice. Considering potential subtypes of PV interneurons in striatum(*Garas et al., 2016*), we compared resting membrane properties including capacitance, membrane potential, tau and holding current between TRN connected and non-connected PV interneurons, and no significant differences were observed (*Figure 3—figure supplement 1a–d*). In order to test the selectivity of this input, neighboring PV-negative cells (putative SPNs) were also recorded, yet none of these neurons exhibited any functional connectivity (0/8 cells) (*Figure 3i*). In current clamp, one second constant blue laser stimulation of TRN axon terminals was sufficient to suppress current injection-evoked spiking in PV interneurons (*Figure 3k*). These results are consistent with the viral tracing data and suggest a selective, functional TRN inhibitory input to striatal PV interneurons.

To determine if the TRN is sufficient to inhibit PV interneurons and affect basal ganglia outputs *in vivo*, optogenetic stimulation of TRN terminals in the striatum was paired with extracellular neuronal recordings using a multi-electrode array (*Figure 3l*, see *Figure 3—figure supplement 2* for array placement) (*Howard et al., 2017*; *Jin et al., 2014*). Of the over two hundred putative SPNs recorded, around 40% show a firing rate change during light stimulation (83/205). Noticeably, half of the responsive SPN population (44/83) exhibited delayed inhibition during the one second constant laser stimulation (*Figure 3m,o*) while others (39/83) exhibited slow latency excitation (*Figure 3n,o*). While the firing modulation observed in SPNs represents the net network effects of TRN stimulation on basal ganglia output, potential disinhibition of SPNs via TRN inhibition of striatal PV interneurons may contribute to some of these effects. Indeed, four putative striatal PV interneurons (fast-spiking interneurons) were identified in our recordings and all of them show inhibited firing activity during

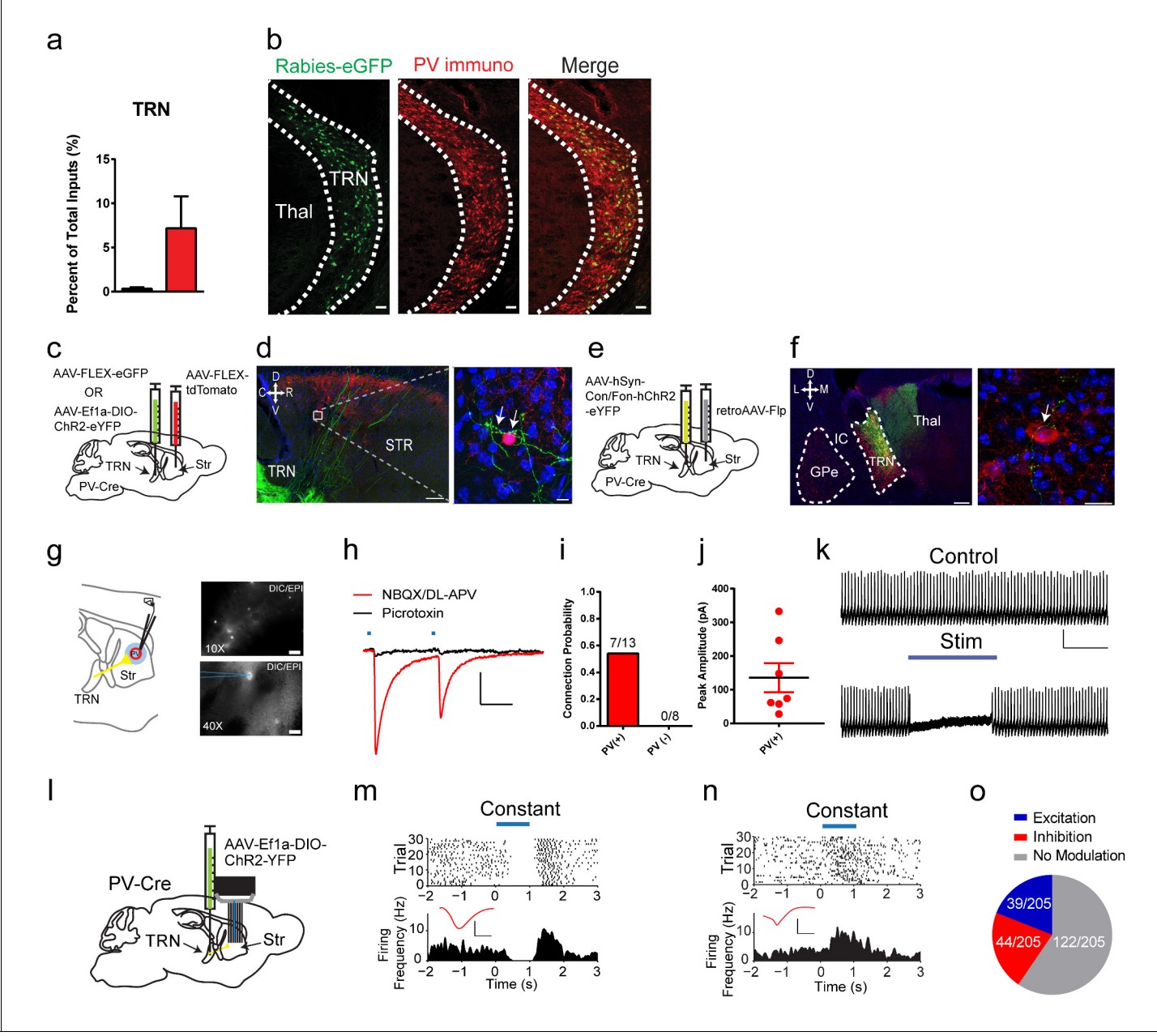

**Figure 3.** A projection from thalamic reticular nucleus to striatal PV interneurons. (a) PV interneurons receive a thalamic reticular nucleus (TRN) projection. (b) Rabies-eGFP positive neurons in TRN colocalize with PV immunostaining. Scale bar, 50 μm. Thal = thalamus. (c) Anterograde viral injection scheme to validate TRN to PV interneuron projection. (d) (left) Cre-dependent AAV eGFP is injected in TRN and Cre-dependent AAV tdTomato injected in the dorsal striatum of a PV-Cre mouse. Scale 200 μm. (right) 63x image of eGFP positive terminals on PV interneuron (red) somas and proximal dendrites (white arrows). Scale 10 μm. (e) A convergent intersectional viral approach to trace TRN projections to striatum. (f) (left) Expression of eYFP in anterior TRN cell bodies and axons seen projecting to dorsal thalamus. Note lack of expression in neighboring GPe. Scale bar, 200 μm. (right) Following PV immunostaining, eYFP fibers (yellow-green) from TRN were observed in close apposition to PV cell somas (red). Scale bar, 20 μm. (g) (left) Slice preparation to functionally validate TRN to PV interneuron projections. (right) Overlays of DIC and red channel epifluorescence at 10x (top) and 40x (bottom). Scale bars 200 μm, 20 μm. (h) Whole-cell recordings of paired light evoked IPSCs (50 ms ISI) following TRN terminal stimulation. IPSCs are blocked with $GABA_A R$ antagonist, picrotoxin (50 μM). Scale bars, 25 ms, 100 pA. PV (+) cells are held at −70 mV. (i) Striatal PV interneurons showed a fast latency, reliable light evoked IPSC, which is absent in the neighboring SPNs. (j) Individual IPSC current amplitudes of connected cells. (k) One second constant blue laser TRN axon stimulation is sufficient to suppress current injection-induced spiking in a PV interneuron. Scale bars, 500 ms, 25 mV. (l) Optogenetic stimulation of TRN terminals in the striatum was paired with extracellular neuronal recordings using a multi-electrode array. (m–n) Exemplar (top) spike raster and (bottom) firing frequency perievent time histogram (PETH) from a SPN exhibiting slow latency light-evoked inhibition (m), or exhibiting slow latency light-evoked excitation (n). Inset (red): Average single unit waveforms of a putative SPN. Scale

*Figure 3 continued on next page*

*Figure 3 continued*

bars 0.2 ms, 50μV. (**o**) Approximately 40% (83/205) of neurons show light-induced firing rate modulation. Half of the responsive SPN population (44/83) exhibited delayed inhibition (red), while others (39/83) exhibited slow latency excitation (blue). See *Figure 3—figure supplement 3* for putative PV interneuron light-evoked responses. The following figure supplements are available for *Figure 3*: *Figure 3—figure supplements 1–3*.

DOI: https://doi.org/10.7554/eLife.35657.008

The following figure supplements are available for figure 3:

**Figure supplement 1.** Membrane properties of striatal PV interneurons are similar between TRN connected or non-connected cells.

DOI: https://doi.org/10.7554/eLife.35657.009

**Figure supplement 2.** Verification of dorsal striatal in vivo recording array placement.

DOI: https://doi.org/10.7554/eLife.35657.010

**Figure supplement 3.** TRN terminal stimulation inhibits putative striatal fast-spiking (PV) interneurons.

DOI: https://doi.org/10.7554/eLife.35657.011

TRN terminal stimulation (*Figure 3—figure supplement 3*). Overall, TRN is capable of functionally regulating striatal PV interneurons and affecting basal ganglia output *in vivo*.

## GPe functional targeting of striatal ChAT and PV interneurons

While previous studies focused on the excitatory inputs to striatum (*Ding et al., 2010*; *Lapper and Bolam, 1992*), inhibition also plays an important role in controlling striatal activity, especially through interneurons (*Koós and Tepper, 1999*; *Silberberg and Bolam, 2015*; *Tepper et al., 2010*). In addition to the novel inhibitory input to striatum from TRN, we explored another major inhibitory projection to the striatum from the GPe (*Bevan et al., 1998*; *Gittis et al., 2014*; *Mallet et al., 2012*; *Saunders et al., 2016*). Rabies tracing revealed that all basal ganglia nuclei similarly innervate both ChAT and PV interneurons, and of these basal ganglia inputs, GPe was the most predominant (two-way ANOVA, $F_{(6, 63)}=27.27$, $p<0.0001$, *Figure 4—figure supplement 1a*). Previous studies have suggested that GPe primarily targets PV interneurons and SPNs (*Bevan et al., 1998*; *Gittis et al., 2014*; *Mallet et al., 2012*; *Saunders et al., 2016*) along with evidence of an anatomical connection to ChAT interneurons (*Guo et al., 2015*; *Mallet et al., 2012*). However, our rabies tracing results found that similar proportions of GPe neurons project to both striatal ChAT and PV interneurons (*Figure 4—figure supplement 1a*). To begin to characterize this GPe projection to ChAT and PV interneurons, rabies-labeled GPe brain sections were immunohistochemically stained for different cell markers (*Hernández et al., 2015*; *Mallet et al., 2016*), and few rabies-eGFP GPe neurons were colocalized with either PV immmunostaining (ChAT-projecting: 3.5%, PV-projecting: 2.8%; *Figure 4—figure supplement 1b–e*) or ChAT immmunostaining (ChAT-projecting: 1.6%, PV-projecting: 1.3%; *Figure 4—figure supplement 1c,d*). Consistent with previous studies, transcription factors NPAS1 and FoxP2, which are enriched in pallidostriatal projection neurons (*Glajch et al., 2016*; *Hernández et al., 2015*), colocalized with a large population of retrogradely labeled eGFP-positive rabies neurons. Rabies eGFP expression for striatal ChAT or PV-projecting GPe neurons colocalized similarly with all immunohistochemical markers tested (*Figure 4—figure supplement 1c*). These data suggest that ChAT and PV interneurons receive comparable numbers of input neurons from potentially overlapping populations of GPe cell types.

To functionally confirm the GPe to striatal connection, pairs of ChAT interneurons and neighboring SPNs were recorded in voltage clamp with brief laser stimulation of ChR2-expressing GPe terminals in striatum. Fast latency, monosynaptic IPSCs were observed in both ChAT-positive (11/11) and ChAT-negative neurons (putative SPNs, 7/7) (*Figure 4a,b*). These IPSCs were confirmed by blockade with the GABA$_A$R antagonist picrotoxin (*Figure 4b*). IPSC amplitudes and probability of release from neighboring ChAT and SPNs were similar (*Figure 4—figure supplement 2a–c*), but SPNs had increased inverse of the coefficient of variation squared ($1/(CV)^2$) (*Figure 4—figure supplement 2d*, Mann Whitney, $p=0.0114$), a presynaptic measure suggesting increased number or function of inhibitory synapses on SPNs. Recordings from pairs of PV interneurons and neighboring SPNs revealed that PV interneurons exhibited larger IPSCs (two-tailed t-test, $p<0.0001$), higher release probability (two-tailed t-test, $p=0.0114$), and greater $1/(CV)^2$ (two-tailed t-test, $p=0.0277$) (*Figure 4b*, *Figure 4—figure supplement 2e–g*), suggesting GPe inhibitory presynaptic release and/or number of synapses is greater on PV interneurons than on SPNs (*Bevan et al., 1998*; *Gittis et al., 2014*; *Glajch et al., 2016*; *Mallet et al., 2012*; *Saunders et al., 2016*). Overall, the ChAT to SPN IPSC ratio was lower

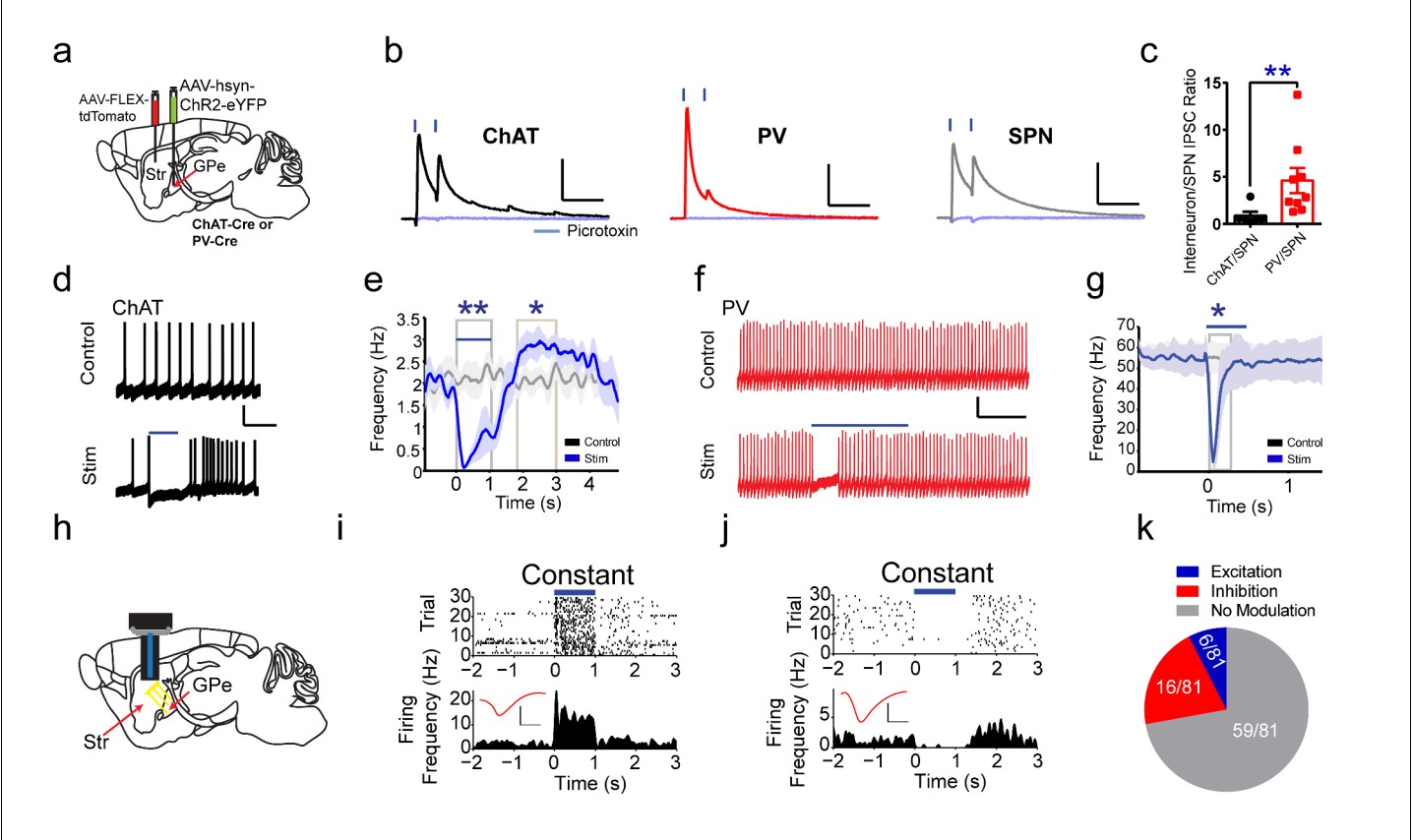

**Figure 4.** Functional projections of GPe to striatal ChAT and PV interneurons. (a) Experimental approach to label GPe inputs with ChR2 and express tdTomato in ChAT or PV interneurons. (b) Averaged paired-pulse (50 ms ISI) light-evoked IPSCs in striatal ChAT interneuron (black), PV interneuron (red) and SPN (grey) following GPe terminal stimulation in striatum. Cells are held at −10 mV in the presence of AMPAR (10 µM CNQX) and NMDAR (50 µM DL-APV) antagonists. IPSCs are blocked by the GABA$_A$R antagonist picrotoxin (100 µM) (light blue trace). Scale bars, 100 ms, 200 pA. See *Figure 4— figure supplement 2* for detailed quantification. (c) The ChAT/SPN paired IPSC ratio is lesser than the PV/SPN paired IPSC ratio (Mann Whitney Test, p=0.0076). (d–e) ChAT interneuron tonic firing is paused by light-evoked stimulation of striatal GPe terminals, evident from single trace (d) and the firing rate PETH (e). Scale bars, 1 s, 20 mV. Dark line represents mean and shading denotes SEM in PETH. GPe terminal stimulation promotes ChAT tonic firing activity pause (two-tailed t-test, p=0.0014) and subsequent rebound (two-tailed t-test, p=0.0469). See *Figure 4—figure supplement 2h–j* for multiple trial raster and quantification. (f–g) Suppression of spiking activity in striatal PV interneurons following GPe terminal stimulation for single trace (f), and the firing rate PETH (g). Scale bars, 250 ms, 25 pA. Quantification of suppression of PV interneuron spiking activity by GPe terminal stimulation (two-tailed t-test, p=0.0401). See *Figure 4—figure supplement 2k,l* for multiple trial raster and quantification. (h) *In vivo* recording of striatal SPNs during optogenetic stimulation of GPe terminals in striatum. (i–j) Representative SPNs showing excitation (i) or inhibition (j) during optogenetic GPe terminal stimulation. Inset (red): Average single unit waveforms of a putative SPN. Scale bars 0.2 ms, 50µV. (k) Pie chart of light modulated SPNs firing activity following GPe laser stimulation. The following figure supplements are available for *Figure 4*: *Figure 4—figure supplements 1* and *2*.

DOI: https://doi.org/10.7554/eLife.35657.012

The following figure supplements are available for figure 4:

**Figure supplement 1.** Cell-type identity of GPe projections to striatal ChAT and PV interneurons.

DOI: https://doi.org/10.7554/eLife.35657.013

**Figure supplement 2.** GPe functional inhibitory connectivity to ChAT interneurons, PV interneurons and SPNs.

DOI: https://doi.org/10.7554/eLife.35657.014

than the PV to SPN ratio, suggesting increased functional connectivity from GPe to PV compared to ChAT interneurons, despite similar numbers of GPe neurons projecting to both (*Figure 4c*, Mann Whitney test, p=0.0076).

Pauses in ChAT interneuron firing are associated with cues that predict reward (*Kimura et al., 1984*), so we next tested if these inhibitory GPe inputs to striatum are capable of suppressing tonic ChAT firing. In whole-cell current clamp recordings, ChAT interneurons were tonically active and rested at modestly depolarized potentials (~ −60 mV, *Figure 4d*). Interleaved blue laser light

stimulation of GPe inhibitory terminals on ChAT interneurons was sufficient to significantly reduce tonic firing (*Figure 4d*, *Figure 4—figure supplement 2h–j*, two-tailed t-test, p=0.0043). After cessation of laser stimulation, ChAT neurons showed significant rebound activity spiking at rates above baseline (*Figure 4e*, *Figure 4—figure supplement 2h–j*, two-tailed t-test, p=0.0469). Furthermore, inhibitory inputs from GPe were sufficient to inhibit current injection-induced spiking in PV interneurons for a few hundred milliseconds following blue laser stimulation (*Figure 4f,g*; *Figure 4—figure supplement 2k,l*; two-tailed t-test, p=0.0401). These data suggest that SPNs, striatal ChAT and PV interneurons all receive inhibitory inputs from GPe, which is sufficient to generate pause-burst activity in ChAT interneurons and suppression of spiking in PV interneurons. To further explore the net functional effects of GPe to striatum projections *in vivo*, extracellular recordings paired with optogenetic stimulation of ChR2-expressing GPe terminals in striatum were performed (*Figure 4h*). It revealed that of all the SPNs responsive to GPe terminal stimulation (22/81), the majority showed inhibited firing activity (16/22) while a small proportion was excited (6/22) (*Figure 4i–k*). These results suggest that GPe sends a prominent inhibitory input to both striatal ChAT and PV interneurons and exert powerful feedback control on striatal activity.

## Excitatory pedunculopontine nucleus projections to striatal ChAT interneurons

The pedunculopontine nucleus (PPN), a part of the ascending reticular activating system, has been observed to project directly to the striatum (*Dautan et al., 2014*); yet, the cell type and physiology of this pathway has not been fully elucidated. Notably, in the rabies tracing data, PPN had a noticeable bias to ChAT interneurons, compared to other subcortical projections (two-way ANOVA, $F_{(1, 72)}$=6.034, p=0.0008). PPN projections to striatum were previously thought to be cholinergic (*Dautan et al., 2014*). However, no overlap in immunostaining for ChAT and eGFP rabies-labeled neurons in the PPN was observed (*Figure 5a*). To begin to characterize this long-range connection, non-Cre dependent AAV-ChR2 was injected into PPN and terminals were observed innervating all basal ganglia nuclei, including striatum (*Figure 5b*). Interestingly, while the eGFP axon terminal expression in the striatum was sparse, it was directed, wrapping the cell somas and proximal dendrites of ChAT interneurons (*Figure 5c*). Overall these data suggest that PPN sends a non-cholinergic projection to striatum that preferentially targets ChAT interneurons.

To functionally explore this input and confirm the anatomical findings, whole-cell voltage clamp recordings of ChAT interneurons combined with optogenetic stimulation of PPN terminals was used to determine connectivity and input type (*Figure 5d–f*). PPN is a heterogeneous hindbrain nucleus composed of cholinergic, glutamatergic and GABAergic neurons (*Martinez-Gonzalez et al., 2011*; *Mena-Segovia et al., 2009*). Recordings at the reversal potential for excitatory and inhibitory transmission before and after the addition of AMPAR/NMDAR antagonists or GABA$_A$R antagonists was used to determine if PPN axon terminals are excitatory or inhibitory, respectively (*Figure 5f*). ChAT interneurons exhibited fast latency, monosynaptic (<6 ms), excitatory currents in a vast majority of neurons sampled (10/11), while only a few neighboring SPNs (3/25) exhibited fast latency (<6 ms), excitatory or inhibitory connections (*Figure 5g*). Notably, the current experiments do not allow us to disambiguate neurotransmitter co-release from a single terminal or differing classes of PPN inputs on a single striatal interneuron (*Wang and Morales, 2009*). Overall, these results suggest that different from SPNs, ChAT interneurons receive strong functional excitatory inputs on the soma and proximal dendrites from PPN.

## Discussion

### Differential excitatory inputs to ChAT and PV striatal interneurons

While some hints at specifying afferent inputs to ChAT and PV interneurons have been observed with anterograde tracers or ultrastructural studies (*Chang and Kita, 1992*; *Gonzales et al., 2013*; *Mallet et al., 2005*; *Mallet et al., 2012*; *Ramanathan et al., 2002*), more comprehensive information on the brain-wide inputs are needed for quantitative comparisons and providing further insights into their functional roles in behavior. Our tracing results suggest that the majority of excitatory inputs to both ChAT and PV interneurons come from the cerebral cortex. Our data confirm previous reports that striatal PV interneurons receive substantial inputs from M1 and S1 (*Parthasarathy and*

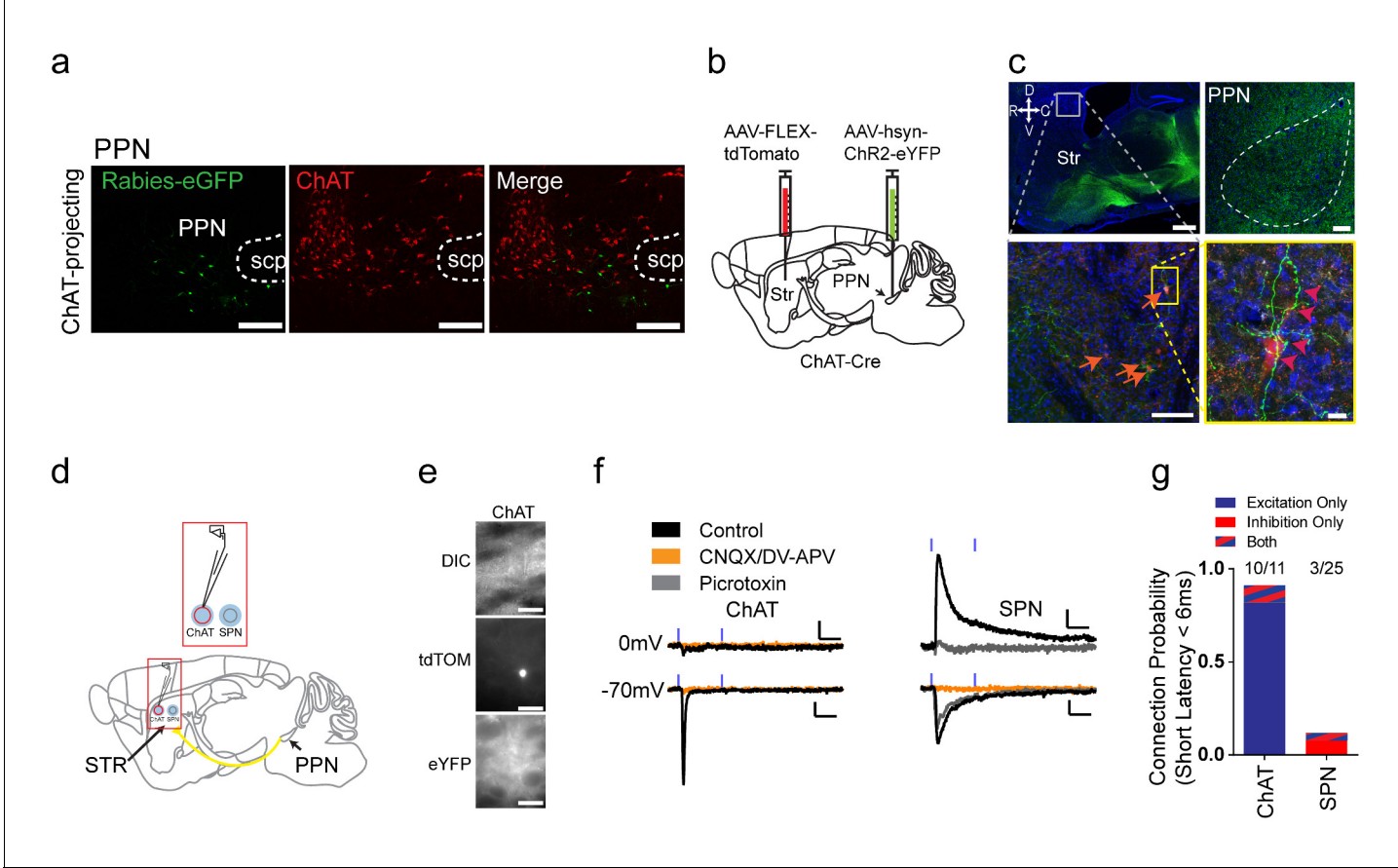

**Figure 5.** Excitatory projections from pedunculopontine nucleus to striatal ChAT interneurons. (**a**) Rabies-labeled projections to striatal ChAT interneurons from the PPN are not cholinergic. Note no colocalized eGFP (green) and ChAT immunostaining (red). scp, superior cerebellar peduncle. Scale bars, 200 μm. (**b**) Viral injection scheme to anterogradely verify the PPN projections to striatum. (**c**) (left) Sagittal image of eYFP terminal expression following injection of AAV-ChR2-eYFP in PPN. Scale bar, 1 mm (top). (right) Sagittal image of PPN injection site with AAV-ChR2-eYFP expression. Scale bar, 100 μm. Note many eYFP fibers from PPN target on tdTomato expressing ChAT interneurons (orange and pink arrowheads). Scale bar, 200 μm (left), 20 μm (right). (**d**) Diagram for functional validation of PPN projections to striatum in brain slices. (**e**) DIC and epifluorescent images of patched dorsal striatal ChAT and neighboring SPN neurons. Scale bar, 200 μm (DIC), 50 μm (epifluorescent). (**f**) Example average current trace at −70 mV or 0 mV under control (black), after blocking AMPAR and NMDAR currents (yellow) or blocking GABA$_A$R currents (grey). Scale bar, 25 ms, 25 pA. (**g**) Percentage of ChAT and SPN neurons exhibiting fast latency (<6 ms) excitatory (blue), inhibitory (red) or both (blue/red stripes) currents following PPN terminal stimulation.

DOI: https://doi.org/10.7554/eLife.35657.015

*Graybiel, 1997*; *Ramanathan et al., 2002*), suggesting a role for PV interneurons in sensorimotor integration. In contrast, we found that projections from associative regions of cortex, including the orbital and cingulate cortex, and the associative regions of the thalamus, including intralaminar nuclei and mediodorsal nucleus, preferentially targeted ChAT interneurons. The orbital cortex has been shown to be involved in decision making, reversal learning and shifting from goal-directed to habit learning (*Gourley et al., 2013*; *Gremel and Costa, 2013*; *McAlonan and Brown, 2003*). The cingulate cortex has been described as an interface between motivation, cognition and action with roles associated with effort, action value, reward expectancy, and behavioral flexibility (*Cowen et al., 2012*; *Hadland et al., 2003*; *Hayden and Platt, 2010*; *Johnston et al., 2007*; *Paus, 2001*; *Shidara and Richmond, 2002*; *Walton et al., 2004*). The thalamic intralaminar nuclei and mediodorsal nucleus are involved in attention and cognitive processing of information (*Kimura et al., 2004*; *Minamimoto and Kimura, 2002*; *Parnaudeau et al., 2013*). This preferential connectivity suggests a specific role of ChAT interneurons in context-dependent modulation of action and behavioral flexibility (*Matamales et al., 2016*). Indeed, recent work suggests cholinergic

interneurons use orbitofrontal inputs to track current environmental state during behavior (*Stalnaker et al., 2016*). Furthermore, we find a novel, functional excitatory pathway from the PPN to striatal ChAT interneurons. This pathway, part of the ascending reticular activating system, may coordinate with the basal forebrain to regulate levels of arousal in cortex and striatum concomitantly, representing a mechanism of context-dependent gain control of behavior (see *Figure 2—figure supplement 2* for visual anatomical summary of ChAT and PV inputs).

When our current data are considered collectively with previous studies using monosynaptic rabies tracing from other striatal cell types (*Smith et al., 2016*; *Wall et al., 2013*), a general theme appears. Each cell type in the striatum appears to have access to the same inputs within a topographic domain (*Hintiryan et al., 2016*), but significant preferences do exist for certain cell types, which imbues functional distinctions. In the case of direct and indirect pathway SPNs, it was found that the amygdala almost exclusively targeted direct pathway, but not indirect pathway SPNs (*Wall et al., 2013*). For the patch/matrix SPNs, the most prominent distinction was limbic subcortical inputs, primarily from the bed nucleus of the stria terminalis preferentially innervating the patch compartment (*Smith et al., 2016*). For ChAT and PV interneurons, our rabies tracing indicates a number of differences that lend insight into the differential role these interneuron subtypes in regulating striatal output to control action selection and learning. Interestingly, when analyzing the cortical laminar breakdown in associative versus sensorimotor cortices we observed more superficial layer projections from associative cortex versus more deep layer projections from sensorimotor cortex in both the ChAT and PV rabies tracing dataset, which was also observed in the patch/matrix rabies tracing data set (*Smith et al., 2016*). This laminar organization may be a general organizing principle of the cortical inputs to the dorsal striatum.

## A TRN to PV interneuron inhibitory circuit

Most previous studies have focused on the excitatory inputs to striatum (*Ding et al., 2008*; *Ding et al., 2010*). Here we report a novel, GABAergic input from the TRN that projects to striatal PV interneurons. This reliable, short latency TRN-PV pathway was sufficient to inhibit PV interneuron spiking and alter SPN activity *in vivo*. However, additional interneuron subtypes will need to be tested to determine the precise mechanism of TRN-mediated SPN modulation. Traditionally, TRN is thought to send projections only within thalamus regulating thalamocortical communications and acting as an attentional spotlight or filter (*Pinault, 2004*). In these models, when one sensory modality is active during attention, a winner-take-all network between thalamus-TRN reciprocal connections would inhibit the other modality and facilitate information processing in the attended domain.

The projection from TRN to striatal PV interneurons found in our current study was first identified through rabies tracing experiments. This projection represents a substantial and selective inhibitory input to striatal PV interneurons as this pathway has not been observed in previous rabies tracing datasets on other striatal cell types (*Guo et al., 2015*; *Smith et al., 2016*; *Wall et al., 2013*). This pathway was further verified independently by anterograde and intersectional tracing experiments in two separate mouse lines along with slice electrophysiology combined with optogenetics.

The function of this TRN-striatal PV interneuron pathway could be particularly significant. Activation of TRN inputs would inhibit PV interneuron activity and potentially excite SPNs via disinhibition. Our electrophysiological recordings from brain slices detected large IPSCs in more than half of striatal PV interneurons sampled. Using *in vivo* striatal recordings with optogenetic stimulation of TRN terminals in striatum, we observed that putative striatal fast spiking interneurons (FSIs) were inhibited during optical stimulation, while a subpopulation of SPNs showed firing rate modulation either directly via PV interneurons or indirectly through a network effect. This subcortical pathway is well positioned to relay attention-related information directly to the striatum. By targeting striatal PV interneurons, we speculate that it might be possible to effectively bias action selection and coordinate an animal's ongoing behavior with corresponding attentional state (*Berke, 2011*; *Gage et al., 2010*; *Jin et al., 2014*). Potentially, this pathway could serve as a subcortical bridge for promptly coordinating sensory attention and actions.

## A GPe to ChAT interneuron inhibitory circuit

Previous data suggested that GPe projected to striatal PV interneurons and SPNs (*Bevan et al., 1998*; *Mallet et al., 2012*) with some anatomical evidence of a projection to ChAT interneurons

(*Guo et al., 2015*; *Mallet et al., 2012*). Our rabies tracing data confirmed these connections as well as demonstrated for the first time a functional connection from GPe to striatal ChAT interneurons. The GPe projections to striatal ChAT and PV interneurons draw from similar proportions of NPAS1, FoxP2 and a small minority of PV and ChAT neurons. However, we did observe a population of rabies-eGFP neurons that did not colocalize with any of the immunohistochemical markers, potentially suggesting another subtype such as Lhx6-expressing GPe neurons (*Mastro et al., 2014*). Utilizing whole-cell recordings, we confirmed that the GPe has a high degree of connectivity to both striatal ChAT and PV interneurons, as well as SPNs. In classical Pavlovian conditioning or stimulus-response learning, ChAT interneurons, which are tonically active, exhibits a pause in firing within a few hundred milliseconds after the presentation of the conditioned stimuli (*Aosaki et al., 1994*; *Kimura et al., 1984*; *Ravel et al., 1999*). This pause response depends on learning and is seen with stimuli predicting both rewarding and aversive outcomes (*Ravel et al., 1999*). While the precise function of the pause response in ChAT interneurons remains to be elucidated at a behavioral level, it has been suggested to encode the salience of external stimuli and regulate local dopamine release in striatum. The neural mechanism underlying the pause has caused an extensive debate in the field. Several cellular mechanisms involving multiple sources have been proposed, including the contributions from the intrinsic cell membrane properties of ChAT interneurons, dopamine regulation or GABA co-release, and thalamic inputs with the involvement of an unidentified type of striatal GABAergic interneuron (*English et al., 2012*; *Nelson et al., 2014*; *Schulz and Reynolds, 2013*; *Straub et al., 2014*; *Sullivan et al., 2008*; *Threlfell et al., 2012*). The abovementioned mechanisms are not necessarily mutually exclusive, and each might actually contribute to the pause response to a different degree or under various conditions.

In our study, we identified a novel extrastriatal GABAergic input from GPe to ChAT interneurons that is sufficient to suppress tonic firing and generate pause responses similar to those observed in behavior (*Aosaki et al., 1994*). Considering recent studies implying that striatum-projecting GPe neurons play a crucial role during action cancelling (*Mallet et al., 2016*), our current results thus further reinforce that the GPe to striatal interneuron projections may be an important part of the feedback circuitry involved in context-dependent action control.

## An excitatory pedunculopontine nucleus input to striatal ChAT interneurons

In this study, we identified a novel, functional excitatory PPN pathway that was selectively enriched in it projections to ChAT interneurons, but not SPNs. Previous literature suggested a significant anatomical cholinergic PPN input to striatum (*Dautan et al., 2014*). Co-release of glutamate from cholinergic terminals in striatum on ChAT interneurons may be one possible source of this excitatory signal. Yet, previous evidence does not lend strong support for the idea of co-release with a vast majority of cholinergic neurons in PPN lacking expression of vGluT2 mRNA (*Wang and Morales, 2009*). Alternatively, the lack of rabies labeling of cholinergic cells in PPN may be attributed to a technical limitation with rabies labeling of neuromodulatory synapses. For example, several striatal rabies tracing studies using the same technique do not find significant rabies labeling of dopamine neurons in the substania nigra pars compacta (SNc) (*Smith et al., 2016*; *Wall et al., 2013*), despite the dense innervation of dopamine terminals from the SNc in dorsal striatum. Alternatively, ChAT interneurons may not receive direct cholinergic projections from PPN. Additional, studies will be needed to address the role of two sources of acetylcholine from ChAT interneurons or PPN terminals in striatum and determine what, if any, crosstalk occurs on striatal interneurons.

Our anatomical tracing data indicate that ChAT interneurons receive more projections from associative regions of cortex and thalamus (e.g. lateral orbital cortex, retrosplenial cortex, mediodorsal thalamus, etc.). Conversely, PV interneurons receive more inputs from sensorimotor regions, including primary motor and somatosensory cortical areas. These preferential inputs suggest a potential functional distinction for their role in controlling behavior, whereby ChAT interneurons may receive internal cognitive or external environmental information for context-dependent action modulation, while PV interneurons may integrate sensorimotor information for action learning and selection. Together, these data indicate a comprehensive revision of striatal circuitry with distinct, yet complimentary roles for striatal ChAT and PV interneurons in shaping striatal function.

# Materials and methods

## Key resources table

| Reagent type (species) or resource | Designation | Source or reference | Identifiers | Additional information |
|---|---|---|---|---|
| Strain, strain background (*Mus musculus*) | ChAT-IRES-Cre | Jackson Labs | stock #006410; RRID:IMSR_JAX:006410 | maintained on a C57BL6/J background |
| Strain, strain background (*Mus musculus*) | PV-Cre | Jackson Labs | stock #008069; RRID:IMSR_JAX:008069 | maintained on a C57BL6/J background |
| Strain, strain background (*Mus musculus*) | Sst-IRES-Cre | Jackson Labs | stock #028864; RRID:IMSR_JAX:028864 | maintained on a C57BL6/J background |
| Strain, strain background (*Adeno-associated virus*) | AAV5-TVA-mCherry | UNC Viral Vector Core | RRID: SCR_002448 | $3–4.3 \times 10^{12}$ particles/mL |
| Strain, strain background (*Adeno-associated virus*) | AAV8-CA-RG | UNC Viral Vector Core | RRID: SCR_002448 | $1.2–4.3 \times 10^{12}$ particles/mL |
| Strain, strain background (*Adeno-associated virus*) | (EnvA) SAD-ΔG Rabies-eGFP | Salk Vector Core | RRID: SCR_014847 | $1.6–6.55 \times 10^{8}$ particles/mL |
| Strain, strain background (*Adeno-associated virus*) | AAV9-Ef1a-DIO-ChR2 (H134R)-eYFP | University of Penn Viral Vector Core | RRID: SCR_015406 | two $\times 10^{12}$ particles/mL |
| Strain, strain background (*Adeno-associated virus*) | AAV9-FLEX-tdTomato | University of Penn Viral Vector Core | RRID: SCR_015406 | two $\times 10^{12}$ particles/mL |
| Strain, strain background (*Adeno-associated virus*) | AAVretro-EF1a-Flp | UNC Viral Vector Core | RRID: SCR_002448 | two $\times 10^{12}$ particles/mL |
| Strain, strain background (*Adeno-associated virus*) | AAV9-hsyn-Con-Fon-hChR2-eYFP | UNC Viral Vector Core | RRID: SCR_002448 | two $\times 10^{12}$ particles/mL |
| Strain, strain background (*Adeno-associated virus*) | AAV9-FLEX-eGFP | University of Penn Viral Vector Core | RRID: SCR_015406 | two $\times 10^{12}$ particles/mL |
| Strain, strain background (*Adeno-associated virus*) | AAV9-hsyn-ChR2-eYFP | University of Penn Viral Vector Core | RRID: SCR_015406 | two $\times 10^{12}$ particles/mL |
| Antibody | anti-PV (mouse,monoclonal) | MilliporeSigma | P3088; RRID: AB_477329 | 1/1000 |
| Antibody | anti-PV (rabbit, polyclonal) | Abcam | ab11427; RRID: AB_298032 | 1/1000 |
| Antibody | anti-ChAT (goat, polyclonal) | MilliporeSigma | AB144P; RRID: AB_2079751 | 1/100 |
| Antibody | anti-NPAS1 (rabbit, polyclonal) | GeneTex | GTX105876; RRID: AB_424768 | 1/500 |
| Antibody | anti-mCherry (mouse, monoclonal) | Takara Bio USA, Inc (Clontech Labs) | 632543; RRID: AB_2307319 | 1/250 |
| Antibody | anti-eGFP (rabbit, polyclonal) | Thermo Fisher Scientific | A11122; RRID: AB_1074875 | 1/400 |
| Antibody | anti-somatostatin (rabbit, polyclonal) | Protos Biotech Corp | NP106SST | 1/300 |
| Antibody | anti-FoxP2 (rabbit, polyclonal) | Sigma | HPA000382; RRID: AB_1078908 | 1/1000 |
| Antibody | donkey anti-mouse 488 (secondary) | Jackson Immunoresearch | 715-545-150; RRID: AB_2340846 | 1/250 |

*Continued on next page*

*Continued*

| Reagent type (species) or resource | Designation | Source or reference | Identifiers | Additional information |
|---|---|---|---|---|
| Antibody | donkey anti-mouse CY3 (secondary) | Jackson Immunoresearch | 715-165-150; RRID: AB_2340813 | 1/250 |
| Antibody | donkey anti-mouse CY5 (secondary) | Jackson Immunoresearch | 715-175-150; RRID: AB_2340819 | 1/250 |
| Antibody | donkey anti-rabbit 488 (secondary) | Jackson Immunoresearch | 711-545-152; RRID: AB_2313584 | 1/250 |
| Antibody | donkey anti-rabbit CY3 (secondary) | Jackson Immunoresearch | 711-165-152; RRID: AB_2307443 | 1/250 |
| Antibody | donkey anti-rabbit CY5 (secondary) | Jackson Immunoresearch | 711-175-152; RRID: AB_2340607 | 1/250 |
| Antibody | donkey anti-goat 488 (secondary) | Jackson Immunoresearch | 705-545-147; RRID: AB_2336933 | 1/250 |
| Antibody | donkey anti-goat CY3 (secondary) | Jackson Immunoresearch | 705-165-147; RRID: AB_2307351 | 1/250 |
| Antibody | donkey anti-goat CY5 (secondary) | Jackson Immunoresearch | 705-175-147; RRID: AB_2340415 | 1/250 |
| Chemical compound, drug | NBQX disodium salt hydrate | MilliporeSigma | N183 | 10 uM (final) |
| Chemical compound, drug | DL-APV | MilliporeSigma | A5282 | 50 uM (final) |
| Chemical compound, drug | Picrotoxin | MilliporeSigma | P1675 | 50–100 uM (final) |
| Chemical compound, drug | NMDG | MilliporeSigma | M2004 | |
| Chemical compound, drug | HCl | MilliporeSigma | H1758 | |
| Chemical compound, drug | KCl | MilliporeSigma | P9541 | |
| Chemical compound, drug | NaH2PO4 | MilliporeSigma | S3139 | |
| Chemical compound, drug | NaHCO3 | MilliporeSigma | S6014 | |
| Chemical compound, drug | Glucose | MilliporeSigma | G5767 | |
| Chemical compound, drug | Sodium L-Ascorbate | MilliporeSigma | A4034 | |
| Chemical compound, drug | Sodium Pyruvate | MilliporeSigma | P2256 | |
| Chemical compound, drug | Thiourea | MilliporeSigma | T8656 | |
| Chemical compound, drug | MgSO4 | MilliporeSigma | M2643 | |
| Chemical compound, drug | CaCl2 | MilliporeSigma | 223506 | |
| Chemical compound, drug | MgCl2 | MilliporeSigma | M9272 | |
| Chemical compound, drug | KMeSO4 | MilliporeSigma | 83000 | |
| Chemical compound, drug | HEPES | MilliporeSigma | H4034 | |
| Chemical compound, drug | EGTA | MilliporeSigma | 3777 | |

*Continued on next page*

*Continued*

| Reagent type (species) or resource | Designation | Source or reference | Identifiers | Additional information |
|---|---|---|---|---|
| Chemical compound, drug | Mg-ATP | MilliporeSigma | A9187 | |
| Chemical compound, drug | Na-GTP | MilliporeSigma | G8877 | |
| Chemical compound, drug | CsCl | MilliporeSigma | C4036 | |
| Chemical compound, drug | CsMeSO3 | MilliporeSigma | C1426 | |
| Chemical compound, drug | QX-314 | MilliporeSigma | L5783 | |
| Chemical compound, drug | TEA-Cl | MilliporeSigma | T2265 | |
| Software, algorithm | MATLAB | | RRID: SCR_001622 | |
| Software, algorithm | GraphPad Prism 6 | | RRID: SCR_002798 | |
| Software, algorithm | Adobe Illustrator CS6 | | RRID: SCR_010279 | |
| Software, algorithm | pClamp9 | | RRID: SCR_011323 | |
| Software, algorithm | Fiji/Imagej | | RRID: SCR_002285 | |
| Other | Allen Reference Atlas | | RRID: SCR_013286 | |

## Delta G-Rabies tracing viral injections

All procedures were approved by the Salk Institute Institutional Animal Care and Use Committee. Group housed male and female adult mice (8–12 weeks) were used in the study. Heterozygous ChAT-IRES-Cre (*Chat*) (Jackson Labs, stock # 006410, RRID:IMSR_JAX:006410) and PV-Cre (*Pvalb*) (Jackson Labs, stock # 008069, RRID:IMSR_JAX:008069) mice were backcrossed to C57Bl6/J (>9 generations). For G-deleted rabies mediated cell tracing, animals were anesthetized with ketamine/xylazine (100 mg/kg/10 mg/kg) and mounted on a stereotaxic device (Kopf Instruments; Tujunga, CA). The skull was leveled at bregma and lambda and a small hole was drilled at the dorsal central striatal coordinate of AP +0.5, ML −1.8. Only the right hemisphere was used for this study. A Hamilton syringe (33 gauge needle) containing freshly mixed AAV5-TVA-mCherry ($3–4.3 \times 10^{12}$ particles/mL; UNC Vector Core; Chapel Hill, NC, RRID: SCR_002448) and AAV8-CA-RG ($1.2–4.3 \times 10^{12}$ particles/mL; UNC Vector Core; Chapel Hill, NC, RRID: SCR_002448) (total 1 µl) was slowly lowered to a depth of DV −2.25 from the dura. The virus cocktail was injected slowly over 10mins. The needle was left in place for 5 min additional minutes and then the needle was slowly retracted over 5 min to reduce virus moving into the tract. Mice were sutured and returned to their home cage with ibuprofen (50 mg/kg/day) in their drinking water for four days. After three weeks to allow for maximal expression of helper viruses animals are injected with 1.5 µl of (EnvA) SAD-ΔG Rabies-eGFP ($1.6–6.55 \times 10^8$ particles/mL, Salk Vector Core, La Jolla, CA, RRID: SCR_014847) on an angle (18°) to avoid labeling any neurons in the initial injection tract in the same target region. Injection locations were identical in ChAT-Cre and PV-Cre animals suggesting that connectivity differences observed are attributable to interneuron target and not topography. A total of (6) ChAT-Cre and (5) PV-Cre mice (high input expression) were used in tracing experiments and included in analysis following a power analysis. These experiments were performed once and all brains (6 ChAT, 5 PV) were included in the analysis.

## Histology and image analysis

Ten days after rabies injection mice were anesthetized with an overdose of ketamine/xylazine and transcardially perfused with 0.01M PBS (30–40 mL) followed by 4% paraformaldehyde (PFA)/0.1M

PB pH 7.4 (30–40 mL) with a peristaltic perfusion pump (Cole Parmer; Vernon Hills, IL). The brain was carefully extracted and post-fixed in 4% PFA overnight (24 hr). The brain was then transferred to 30% sucrose/0.1M PB for 1–2 days until the brain equilibrated and sunk. The brain was then blocked with a brain matrix (Zivic Instruments; Pittsburg, PA) to obtain a true coronal plane and mounted on a freezing microtome. Additional ChAT-Cre and PV-Cre animals were cut in the sagittal plane to avoid missing any targets in rostral or caudal sites. Only neurons ipsilateral to the injection site were quantified. Coronal slices were collected at 50 µm resolution in 96 well plates containing cyroprotectant (0.1M phosphate buffer, ethylene glycol, glycerol) to maintain AP position. Every other brain slice was plated on super frost plus slides (Thermo Fisher Scientific, Waltham, MA) for a whole brain reconstruction at 100 µm resolution. Slides were counterstained with DAPI and cover slipped with aqua-poly mount mounting media (Polysciences, Inc; Warrington, PA). Slides were scanned on an automated slide scanner (Olympus VS120) at 10x in the blue and green channels. Images were batch converted to composite TIFFs and saved for image analysis. Individual channels were thresholded when necessary to better distinguish cells in densely expressing regions. Coronal mouse brain reference atlas images overlays were used from the Allen Mouse Brain Atlas (RRID: SCR_013286) (Website: © 2015 Allen Institute for Brain Science. Allen Mouse Brain Atlas [Internet]. Available from: http://mouse.brain-map.org) and made into to transparent overlays in Adobe Illustrator CS6 (San Jose, CA)(RRID: SCR_010279). De-identified images were placed in Adobe Illustrator CS6 and the proper AP coronal reference image was overlaid on the scanned brain slice. The reference image was warped to match slice boundaries and white matter anatomical landmarks at multiple points. Each cell soma in each brain region and cortical layer was counted and AP plane was recorded. Counters were blind to condition and were vetted with an inter-rater reliability of >95% when counting the same brain before analysis. To check the most rostral and caudal regions of the brain for expression additional mouse brains in the sagittal plane were analyzed. Total number of neurons minus striatal expression was used to determine the total number of input neurons to either ChAT or PV interneurons. All data is presented as the percent of total number of input neurons to normalize for difference in the number of neurons labeled from mouse to mouse.

Higher order association cortex grouping included orbital, prelimbic, insular, cingulate, secondary motor cortex, secondary somatosensory cortex, retrosplenial and posterior parietal cortex. Sensorimotor cortex grouping included primary motor cortex, primary somatosensory cortex as well as sensory regions like auditory cortex and visual cortex. Cortical layer distribution was identified by overlaid brain atlas template aligned to corpus callosum and edge of cortex along with DAPI counter staining. Distinctive layer 4 DAPI staining was also used as a landmark to divide superficial or deep layer expression in certain cortical regions.

For starter cell quantification six slices containing the rostral to caudal striatum and surrounding the injections site were immunostained with anti-mCherry (TVA) and anti-eGFP antibodies. Each image was bisected into dorsal medial and dorsal lateral striatum subdivisions. Total number of TVA-mCherry and rabies-eGFP colocalized starter cells were counted in image j. Data is presented as percent of total starter cells.

## Immunohistochemistry

Rabies-eGFP brain slices were washed 3x in 1X TBS (15 min/each) in net wells on an orbital shaker. Slices were blocked and permeabilized in TBS++ (3% normal horse serum (Jackson ImmunoResearch, West Grove, PA), 0.25% TritonX-100 in 1X TBS) for 1 hr at room temperature on a shaker. Primary antibodies (anti-PV mouse, 1:1000, P3088 (MilliporeSigma, St. Louis, MO; RRID: AB_477329); anti-PV rabbit, 1:1000, ab11427 (Abcam, Eugene, OR; RRID: AB_298032); anti-ChAT goat, 1:100, AB144P (MilliporeSigma, St. Louis, MO; RRID: AB_2079751); anti-NPAS1 rabbit, 1:500, GTX105876 (GeneTex; Irvine, CA; RRID: AB_424768); anti-mCherry mouse 1:100, 632543 (Takara Bio USA, Inc, Mountain View, CA; RRID: AB_2307319); anti-eGFP rabbit, 1:400, A11122 (Thermo Fisher Scientific, Waltham, MA; RRID: AB_1074875); anti-Somatostatin rabbit 1:300, NP106SST, (Protos Biotech Corp, New York, NY); anti-FoxP2 rabbit 1:1000, HPA000382, (Sigma; RRID: AB_1078908)) was diluted in TBS ++and incubated on a shaker at 4°C for 48 hr. Slices are washed 2x in 1X TBS (15 min/each) and 30 min in TBS++. Slices were then incubated in secondary antibody (donkey anti-mouse (RRID: AB_2340846, RRID: AB_2340813, RRID: AB_2340819) or donkey anti-rabbit (RRID: AB_2313584, RRID: AB_2307443, RRID: AB_2340607) or donkey anti-goat (RRID: AB_2336933, RRID: AB_2307351, RRID: AB_2340415) Cy2, Cy3 or Cy5, 1:250, (Jackson

ImmunoResearch, West Grove, PA) for 2–3 hr at room temperature on an orbital shaker. Slices were washed three times in 1XTBS (15 min/each), transferred to 0.1M PB, plated on super frost plus slides and cover slipped with aquapoly mount mounting media. Images were collected on a Zeiss 780 confocal microscope with tiling capabilities. Orthogonal projections of the Z-plane of individual channels are presented. Colocalization analysis is performed in Fiji/ImageJ (RRID: SCR_002285). Briefly, individual channels are opened in Fiji/ImageJ and merged into a composite. The area of interest is isolated, brightness and contrast adjusted and individual channels are toggled on and off to determine colocalization with the cell counter tool. Total numbers of neurons in both channels are counted over multiple slices in the A-P axis using the cell counter tool and the number of neurons colocalized is reported as a percent of total eGFP neurons.

## Surgeries for functional validation of connectivity

*TRN*: PV-Cre mice between 10–16 weeks were anesthetized and placed in a stereotaxic frame. The skull was leveled at bregma and lambda and a mounted drill was used to create holes in the skull over the TRN and dorsal striatum based off bregma. A Hamilton syringe (33 gauge needle) was slowly lowered to the TRN (AP −0.58, ML +−1.25, DV −3.5) and 0.1 µl of AAV9-Ef1a-DIO-ChR2 (H134R)-eYFP (titer ~2×10^{12}, Penn Vector Core; RRID: SCR_015406) was injected over 10 min. The needle was left in place for 5 min. With a different syringe 1 µl of AAV9-FLEX-tdTomato (titer ~2×10^{12}, UNC Vector Core; RRID: SCR_002448) was injected into the dorsal striatum (AP +0.5, ML +−1.9, DV-2.25) to label PV interneurons in a PV-Cre mouse. The AAV9-FLEX-tdTomato was never observed in TRN. In an additional approach to validate TRN connectivity 0.5 µl of AAVretro-EF1a-Flp (UNC Vector Core; RRID: SCR_002448) is injected in dorsal striatum (AP + 0.5, ML +−1.9, DV-2.25) and during the same surgery 0.1 µl of AAV9-hsyn-Con-Fon-hChR2-eYFP (UNC Vector Core; RRID: SCR_002448) is injected in TRN (AP −0.58, ML +−1.25, DV −3.5). Additionally, Sst-IRES-Cre (*Sst*) (Jackson stock # 028864, RRID:IMSR_JAX:028864) mice were injected with AAV9-FLEX-eGFP in the TRN like previous. *GPe:* For GPe to ChAT and PV interneuron recordings 0.2 µl of AAV9-hsyn-ChR2-eYFP (Penn Vector Core; RRID: SCR_015406) was injected in the GPe (AP −0.46, ML +−1.9, DV −3.3) similar to above. 1 µl of AAV9-FLEX-tdTomato (titer ~2×10^{12}, UNC Vector Core; RRID: SCR_002448) was injected into the dorsal striatum (AP +0.5, ML +−1.8, DV-2.25) of a ChAT-Cre or PV-Cre mouse to identify ChAT and PV interneurons, respectively. *PPN:* For validation of PPN to striatal interneuron connectivity we injected a small volume (0.1 µl) of AAV9-hsyn-ChR2-eYFP in PPN (AP −4.48, ML +−1.1, DV −3.2) (Penn Vector Core; RRID: SCR_015406) and then injected 1 µl of AAV9-FLEX-tdTomato (AP +0.5, ML +−1.9, DV −2.25) (UNC Vector Core; RRID: SCR_002448) in the striatum in a ChAT-Cre or PV-Cre animal to identify ChAT and PV interneurons.

## Ex vivo brain slice electrophysiology

Two weeks following AAV injection, mice are anesthetized with ketamine/xylazine and transcardially perfused with ice cold, bubbling (95% $O_2$/5% $CO_2$) NMDG cutting solution [consisting of (in mM): NMDG 105, HCl 105, KCl 2.5, $NaH_2PO_4$ 1.2, $NaHCO_3$ 26, Glucose 25, Sodium L-Ascorbate 5, Sodium Pyruvate 3, Thiourea 2, $MgSO_4$ 10, $CaCl_2$ 0.5, 300mOsm, pH = 7.4]. The brain is blocked coronally or sagittally with a brain matrix (Zivic Instruments; Pittsburg, PA) and acute coronal/sagittal slices (300 µm) were cut on a vibratome (VT1000S, Leica Microsystems; Buffalo Grove, IL) through the striatum in ice cold, bubbling NMDG based cutting solution. Slices were allowed to recover for 15 min at 32°C in bubbling NMDG cutting solution. Slices were then transferred to a holding chamber consisting of normal ACSF [consisting of (in mM): NaCl 125, KCl 2.5, $NaH_2PO_4$ 1.25, $NaHCO_3$ 25, D-Glucose 12.5, $MgCl_2$ 1, $CaCl_2$ 2, pH = 7.4, 295 mOsm] at 28°C. After at least one hour of recovery the slices were placed in a chamber and aCSF was perfused over the slices at ~2 mL/min. PV (+) cells were visualized under IR- DIC optics (Ziess Axioskop2; Oberkocken, Germany) at 40x and confirmed to express tdTomato with brief confirmation in the epifluorescent channel. In some experiments neighboring PV (-) neurons (putative SPNs) were patched. Voltage clamp (28°C) or current clamp (33°C) recordings were made from PV-tdTomato (+) interneurons in the dorsal striatum (~+0.5 mm bregma). 3–4 MΩ patch pipettes (WPI; Sarasota, FL) were pulled from borosilicate glass on a P-97 pipette puller (Sutter Instruments; Novato, CA) and filled with internal solution consisting of (in mM): $KMeSO_4$ 135, KCl 5, $CaCl_2$ 0.5, HEPES 5, EGTA 5, Mg-ATP 2, Na-GTP 0.3, pH = 7.3, 305mOsm) for current clamp studies. For PV voltage clamp studies to examining TRN connectivity

IPSCs were isolated with a high Cl- based internal solution consisting of (in mM): CsCl 120, CsMeSO$_3$ 15, NaCl 8, HEPES 10, EGTA 0.5, QX-314 5, Mg-ATP 2, Na-GTP 0.3, pH 7.3, 305mOsm. Following five minutes post break in paired light pulses (473 nm, 5–25 mW/mm$^2$, 2.5 ms, 50 ms ISI) were delivered through a 200 μm glass fiber optic (Thor Labs; Newton, NJ) positioned close to the recorded cell (50–150 μm) at 0.05 Hz using a 473 nm blue DPSS laser system (Laserglow Technologies, Toronto, ON). Twenty sweeps (0.05 Hz) were collected to determine latency and CV. The cell was held at −70 mV and light evoked currents are collected after bath application of 10 μM NBQX and 50 μM DL-APV (MilliporeSigma, St. Louis, MO) to block AMPAR and NMDAR-mediated transmission, respectively. IPSCs were collected 8–10 min following the wash in of drugs. 50 μM-100μM picrotoxin (MilliporeSigma, St. Louis, MO) was added to the bath to block fast GABA$_A$R transmission and confirm IPSC. Series resistance was initially compensated and monitored continuously throughout the experiment and the data were rejected if the parameters changed by more than 20% over the duration of the recording.

For ChAT, PV and SPN IPSC voltage clamp studies Cs+ methanesulfonate internal solution consisting of (in mM): CsMeSO3 (120), NaCl (5), TEA-Cl (10), HEPES (10), QX-314 (5) EGTA (1.1), Mg-ATP (4), Na-GTP (0.3), pH = 7.2–7.3, 305mOsm paired with holding the cell at −10 mV. Light evoked (473 nm, 5–25 mW/mm$^2$, 2.5 ms, 50 ms ISI) IPSCs are collected in the presence of 10 μM NBQX and 50 μM DL-APV to block AMPAR and NMDAR-mediated transmission, respectively. 50 μM-100μM picrotoxin was added to the bath to block fast GABA$_A$R transmission and confirm IPSCs. Pairs of ChAT-Cre tdTomato positive or PV-Cre tdTomato positive and neighboring tdTomato negative putative SPNs (<100 μm) are recorded from the same slice. For PPN connectivity experiments Cs +-methanesulfonate internal is used like above. Pairs of ChAT and SPN (<100 um) are used for connectivity probability. A cell is considered connected if it has a visible, reliable current (20 sweeps, 0.05 Hz) with onset latency less than 6 ms post laser on. Excitatory and inhibitory currents are determined following wash on of AMPAR/NMDAR antagonists and GABA$_A$R antagonists, respectively. Voltage-clamp recordings were performed using a Multiclamp 700A (Molecular Devices; Sunnyvale, CA), digitized (Digidata 1440; Molecular Devices; Sunnyvale, CA) at 10 kHz and filtered at 2 kHz.

For GPe current clamp studies a potassium methane sulfonate internal solution is used. To avoid run down of spiking in ChAT interneuron in whole cell current clamp mode recordings were collected in the first 15 min following break in or in loose patch configuration. Three second threshold current injections (+150–350 pA) were given to drive the PV neuron to fire consistent APs. On interleaved trials 1 s constant or 20 Hz blue light stimulation (473 nm, 5–10 mW/mm$^2$) was given to activate GPe ChR2 (+) fibers in the striatum and spiking inhibition was quantified over many sweeps. Perievent time histograms (PETH) are constructed, aligned to laser on and smoothed in MATLAB (Natick, MA). Average spiking during one second stimulation or control (no laser) are compared. Rebound window (1.8–2.9 s average following laser on) is used for rebound statistics. Current clamp recordings were filtered and digitized at 10 kHz and collected with pClamp 9 (Molecular Devices; Sunnyvale, CA; RRID: SCR_011323). Data was analyzed with Clampfit nine and custom MATLAB (RRID: SCR_001622) scripts (see source code MATLAB file).

## In vivo electrophysiology

Striatal neurons were recorded as previously described (Jin et al., 2014). Briefly, mice were lightly anesthetized using isoflourane (4% induction; 1–2% sustained) and were placed in a stereotactic frame. For electrophysiological recording, we utilized electrode arrays (Innovative Neurophysiology Inc.; Durham, NC) of 16 tungsten contacts (2 × 8) that were 35 μm in diameter. Electrodes were spaced 150 μm apart in the same row and 200 μm apart between two rows. Total length of electrodes was 5 mm. Each array with an optic fiber directly attached was employed. The tip of the fiber was ~200 μm away from the tips of the electrodes and the optic fiber was firmly fixed to the array for the duration of each recording session. Array targeting dorsal striatum (+0.5 AP,±1.5 ML, -2.0 ~ 2.2 DV) was incrementally lowered into dorsal striatum. Silver grounding wire was attached to skull screws.

Neural activity was recorded using the MAP system (Plexon Inc.; Dallas, TX). The spike activities were initially online sorted with a built-in algorithm (Plexon Inc., Dallas, TX). Only spikes with stereotypical waveforms clearly distinguished from noise and with relatively high signal-to-noise ratio were tagged and saved for further analysis. After the recording session, the recorded spikes were further isolated into individual units by an offline sorting software (Offline Sorter, Plexon Inc, Dallas, TX).

Each individual unit displayed a clear refractory period in the inter-spike interval histogram, with no spikes during the refractory period (larger than 1.3 ms).

To optogenetically stimulate GPe or TRN terminals within striatum, we injected non-floxed version of AAV-ChR2 virus (0.2 µl of AAV9-hsyn-ChR2-eYFP, Penn Vector Core) into the GPe (AP −0.46, ML + −1.9, DV −3.3) or a Cre-dependent AAV-ChR2 in a PV-Cre mouse for TRN (AP −0.58, ML + −1.25, DV −3.5) (0.1 µl of AAV9-Ef1a-DIO-ChR2(H134R)-eYFP, Penn Vector Core) (titer ~$2\times10^{12}$). For each recording session, blue laser stimulation was delivered through the optic fiber from a 473 nm laser (Laserglow Technologies, Toronto, ON) via a fiber-optic patch cord, and the neuronal responses were simultaneously recorded. The stimulation patterns included 1 s constant light and 20 or 50 Hz (10 ms pulse width, 20 or 50 pulses in 1 s). The inter-stimulation interval was 4 s and each stimulation pattern was repeated for 30 trials. The laser power was adjusted carefully (~3.0–5.0 mW) to drive reliable response.

## Statistics

Statistics were conducted in Graph Pad Prism 6.01 (La Jolla, CA; RRID: SCR_002798). Student two-tailed t-test or non-parametric Mann Whitney U Test was conducted when distributions significantly deviated from normal distributions. Two-way ANOVA with Sidak's multiple comparisons correction was used to analyze whole brain input regions and cortical layer distributions.

## Acknowledgements

The authors would like to thank Ed Callaway, Rui Costa, Tom Jessell, Chris Kintner, Saket Navlakha and members of the Jin lab for discussion and comments on the manuscript. This work was supported by grants from the NIH (R01NS083815 and R01AG047669), the Dana Foundation, the Ellison Medical Foundation, and the Whitehall Foundation to XJ.

## Additional information

### Funding

| Funder | Grant reference number | Author |
|---|---|---|
| National Institutes of Health | R01NS083815 | Xin Jin |
| National Institutes of Health | R01AG047669 | Xin Jin |
| Dana Foundation | | Xin Jin |
| Ellison Medical Foundation | | Xin Jin |
| Whitehall Foundation | | Xin Jin |

The funders had no role in study design, data collection and interpretation, or the decision to submit the work for publication.

### Author contributions

Jason R Klug, Conceptualization, Data curation, Formal analysis, Supervision, Validation, Investigation, Visualization, Methodology, Writing—original draft, Project administration, Writing—review and editing, Performed rabies tracing, viral injections, histology, immunohistochemistry, microscopy, and slice electrophysiology, Analyzed data and wrote the manuscript; Max D Engelhardt, Formal analysis, Conducted input cell counting and assisted in viral injections and histology; Cara N Cadman, Formal analysis, Conducted input cell counting; Hao Li, Data curation, Formal analysis, Performed the in vivo electrophysiology and analyzed the data; Jared B Smith, Data curation, Writing—review and editing, Conducted an intersectional AAV tracing experiment and helped edit the manuscript text; Sarah Ayala, Formal analysis, Conducted the input cell counting; Elora W Williams, Data curation, Conducted starter cell counts; Hilary Hoffman, Data curation, Assisted in the viral injections and histology; Xin Jin, Conceptualization, Resources, Supervision, Funding acquisition, Writing—original draft, Project administration, Writing—review and editing

## Author ORCIDs

Jason R Klug (ID) https://orcid.org/0000-0002-1774-6007

Xin Jin (ID) http://orcid.org/0000-0002-1106-4013

## Ethics

Animal experimentation: This study was performed in strict accordance with the Guide for the Care and Use of Laboratory Animals of the Salk Institute for Biological Studies. All of the animals were handled according to approved institutional animal care and use committee (IACUC) protocols (#12-00032) of the Salk Institute for Biological Studies. All surgery was performed under isoflurane or ketamine/xylazine anesthesia, and every effort was made to minimize suffering.

## Decision letter and Author response

Decision letter https://doi.org/10.7554/eLife.35657.019

Author response https://doi.org/10.7554/eLife.35657.020

# Additional files

## Supplementary files

• Source code 1. MATLAB script for perievent time histogram (PETH) plot. Plot PETH of ChAT or PV interneuron firing rates aligned to laser on (time zero).

DOI: https://doi.org/10.7554/eLife.35657.016

• Transparent reporting form

DOI: https://doi.org/10.7554/eLife.35657.017

## Data availability

All data generated or analyzed during this study are included in the manuscript and supporting files.

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
