## [Decision Letter]

Thank you for submitting your article "Differential inputs to striatal cholinergic and parvalbumin interneurons manifest functional distinctions" for consideration by *eLife*. Your article has been reviewed by three peer reviewers, and the evaluation has been overseen by a Reviewing Editor and a Senior Editor. The following individuals involved in review of your submission have agreed to reveal their identity: Thomas A Stalnaker (Reviewer #1); Jesus Bertran-Gonzalez (Reviewer #3).

The reviewers have discussed the reviews with one another and the Reviewing Editor has drafted this decision to help you prepare a revised submission.

Summary:

This manuscript utilizes an array of techniques including viral tracing, genetics, and electrophysiology to characterize the inputs to and potential functional roles of the parvalbumin and cholinergic striatal interneurons. Identifying the distinct circuits and potential roles of these two striatal compartments is an important and long-standing question in systems neuroscience, and the current experiments were judged to be a valuable addition to this work.

Essential revisions:

While all three reviewers appreciated the sheer amount and also the high quality of much of the data, they also felt that overall the paper came across as somewhat inconclusive or unfocused. On discussion, it was generally agreed that this was due to the lack of consistency in the behavioral approaches and the results. In contrast to the anatomical and ex vivo electrophysiological data, the behavioral experiments tended to suffer from major weaknesses (inconsistent designs, lack of opto controls, use of primarily excitatory methods, non-physiologic parameters) that made their interpretation difficult and/or there were inconsistencies across experiments that were difficult to resolve. These issues are evident across the reviews. While the authors may feel that these can all be resolved, the consensus of the reviewers was that the paper would be much better if it were refocused on the anatomical comparisons, with the behavioral results removed. This change, along with some adjustments to the comparisons (point #2, reviewer #3), were judged to be the essential revisions. We realize this is a large change, but if the authors are willing to do this, the reviewers felt a revision would be worthwhile.

*Reviewer #1:*

This manuscript presents a wide array of experiments whose overarching object is to characterize the inputs to and functional roles of parvalbumin (PV) and cholnergic (CIN) striatal interneurons.

The experiments are technically sophisticated and the results are very thoroughly and clearly reported. There is a veritable mountain of interesting data in this manuscript. The anatomical tracing results are especially valuable and important. The one general weakness is that the functional experimental results are suggestive rather than constituting solid tests of the hypotheses in question. Below, I will try to specify what I mean. However, in spite of this weakness, I think that the data taken together, particularly the anatomical data, are of such importance and so thoroughly presented that the manuscript should be published with only minor revisions.

1) I don't think that the functional data regarding the thalamic reticular nucleus inputs to PV neurons say very much. Here the authors stimulated these inputs in freely moving mice and found that locomotion was increased. However, there is no way to tell what this might mean. In the Discussion, this result is linked to a role for this input in modulation of attention, biasing action selection, and coordinating sensory attention and action. But really the result does not provide evidence supporting any of those ideas one way or the other.

2) The external segment of the globus pallidus inputs onto CINs are suggested to be involved in generating CIN pauses and possibly in reinforcement learning. But pause-like responses occur when these inputs are stimulated continuously for nearly a second, which is unlikely to be a physiologically relevant mechanism. The authors also show that mice will learn to push a lever to stimulate these inputs and in a separate experiment to inhibit CINs. These results are interesting and suggestive. However, the latter effect is only about one tenth the magnitude of the former effect, suggesting that some other mechanism besides inhibiting CINs must be in play. I think the authors should comment on the disparity between the magnitude of these two results. What other mechanism (besides inhibition of CINs) might account for the highly robust self-stimulation of these inputs?

3) The behavioral effects of ablating CINs or PV neurons are suggestive and interesting, but again I think the conclusions reached are far from clearly supported. The PV-ablation effect on operant learning is very convincing. However, this does not seem consistent with what others have suggested in regard to PV neurons – e.g. a role in action selection. Can the authors clarify whether their data shed any light on why PV-ablation results in such an enormous learning deficit? In the case of the CIN-ablation effect on the go/no go task, can the authors shed any additional light on why they interpret this as a deficit in context-dependent control of actions? Under this interpretation, why would the effect on baseline responding appear on day 1, before any learning of context has occurred in control mice? Wouldn't an alternative interpretation be that these rats have an unconditioned increase in food-elicited hyper-activity that they slowly learn to suppress?

Finally, two other questions/issues that I think the authors should address:

a) They should clarify how many mice contributed to the rabies tracing experiment (sorry if I missed this), and where the placements were located. How did the authors confirm that the initial infusions of virus took place in identical locations in the PV-cre vs. Chat-cre mice? Furthermore, are there any data in this experiment speaking to whether associative cortical inputs to CINs occur more in the dorsomedial locations in striatum or sensorimotor inputs to PV neurons occur more in dorsolateral locations? The authors show that overall the number of originator neurons in dorsomedial vs. dorsolateral are similar, but doesn't that proportion vary in each brain, and if so, wouldn't that shed some light on my question?

b) Can the authors comment on why they might have found that the pedunculopontine inputs to striatum are not cholinergic, in contrast to previously published data?

*Reviewer #2:*

In this manuscript Klug et al. use viral tracing, genetics, and electrophysiology to map the inputs to parvalbumin (PV) and cholinergic (ChAT) interneurons in the dorsal striatum. They identify some previously unknown or underappreciated inputs, particularly the TRN input to PV interneurons, which will be of interest to the basal ganglia field. I found the input tracing part of their study (Figures 1, 2, 3, 6, and accompanying supplementary figures) compelling, novel, and believe it deserves to be published in *eLife*.

The authors also use optogenetics to examine the function of three prominent inputs to PV or ChAT cells: a previously unknown GABAergic projection from the thalamic reticular nucleus, a projection from GPe, and a projection from the pedunculopontine nucleus. While I applaud the authors' intentions and recognize the huge effort that went into these experiments, I felt that these behavioral experiments were not well designed because there were some big inconsistencies in behavioral tasks, and the controls were sometimes not defined or poorly chosen. Additionally, in my reading the cell ablation study at the end seems to contradict some of the optogenetics results. For example, in Figure 4 they show that activating the GABAergic TRN projection to striatum-which preferentially targets PV cells-increases distance traveled. But in Figure 7 they show that genetically ablating PV cells doesn't affect distance traveled but it does affect time in center of arena (a measure of anxiety). Not only do these results seem contradictory, but the data the authors choose to plot in Figures 4 and 7 are inherently different. I was also confused why the authors used an ICSS task in Figure 5, but a food reward task in Figure 7. I also had a hard time finding information about number of animals or how the control experiments were designed. There were also few if any GFP or YFP control experiments for the optogenetics part. Finally, the authors over-relied on optogenetic excitation of different inputs, which raises the concern about showing sufficiency not necessity.

I don't think the manuscript in its current form can be published without major changes to the behavioral experiments, or elimination of the behavioral experiments section altogether and publication of just the input tracing part.

*Reviewer #3:*

This study gathers an impressive range of techniques and presents a comprehensive anatomical, physiological and behavioural dataset aimed at determining the specific circuitry controlling cholinergic (ChAT) and parvalbumin (PV) interneurons of the striatum. The authors start by identifying, at the whole-brain scale, the immediate circuitry inputting each interneuronal system through retrograde trans-synaptic tracing, which then use to confirm some of the suggested circuitry and, more importantly, identify previously unrecognised projections. The authors go on and characterise three of these new circuits (TRN to PV, GPe to ChAT and PPN to ChAT) and also conduct confirmatory behavioural experiments whereby each interneuronal subpopulation is selectively ablated using toxicogenetics. Overall this study is excellent and will significantly contribute to this field of research.

I have however a few observations that would be worth clarifying/discussing before this manuscript reaches its final form:

1) The finding that the same number of starter cells are present in the two Cre lines is puzzling (Figure 1—figure supplement 2E-G). Does this mean that the actual density is the same in both populations? This contrasts with the view that PV neurons are more frequent than ChAT neurons in the dorsal striatum. Perhaps it would be worth contrasting those numbers with those obtained in WT mice, just to rule out that Cre expression does not limit the proliferation of any of the two populations, as seems to be happening in certain lines (e.g. Harno et al. Cell Metabolism, 2013, 18 (1); 21-28). This is only a suggestion that might help reconciling with previous literature.

2) In many instances in the paper, the percent of total inputs to PV and ChAT interneurons is statistically compared across the two systems side by side (e.g. Figure 1—figure supplement 2F-G; Figure 2B-F; Figure 3A-C; Figure S4A-L; Figure 4—figure supplement 1A). I am unsure about the validity of this comparison. To me, even if the number of starter neurons is similar and the area is matching, these two subpopulations form entirely different systems with very dissimilar physiological and morphological characteristics (e.g. Kawaguchi, JNeurosci, 1993,13(11); 4908-23). Features like dendritic arborisation, synaptic density and overall cellular volume may influence to a large extent the efficiency of trans-synaptic infection, therefore providing a very different pattern of labelling in circuits inputting onto PV and ChAT neurons. While this provides very valuable information on the relative proportion of afferent projections from diverse brain regions to each interneuronal system separately, I fail to see how comparing across the two systems is informative. Thus, claims such "ChAT neurons received significantly more inputs from the associative cortex than PV neurons" would be misleading. I think that all data should be presented (and statistical comparisons done) in each system separately-comparisons like "ChAT neurons received significantly more inputs from associative than sensorimotor cortices whereas PV neurons showed the opposite trend" would be appropriate. This would also reduce the amount of statistical comparisons, and would allow using within-subjects analyses which can be more powerful.

3) In the TRN study (subsection “Different thalamic projections to striatal ChAT vs. PV interneurons”, second paragraph), it is not immediately evident that parvalbumin and somatostatin staining inform about the anatomical boundary of the TRN, while this is a critical point in later viral manipulations (i.e. Figure 3A). It would be helpful to clarify this a bit more in the text, perhaps providing citations that describe the PV and SOM markers in TRN.

4) While the full characterisation of the transgenic lines presented at the very start of the paper is appreciated, I find that some viral spread and/or multielectrode placement confirmation may be lacking in some of the in vivo experiments. For example the TRN and GPe multielectrode/optogenetics experiments (Figure 4, Figure 3—figure supplement 3, Figure 5, Figure 4—figure supplement 2) do not have placement verifications. Adding a diagram with the locations in each experimental animal could be helpful.

5) The toxicogenetics experiments seem very clean and are a very good option to expose the role of each interneuronal system in action control. In the case of ChAT ablation, the authors did not find any deficit in the initial acquisition of instrumental behaviours (Figure 7), and they found an increase of baseline lever-pressing rate which had no consequence on later stimulus-response learning. The authors argue that these data support that ChAT interneurons play a role in providing context to the modulation of action. However, I think that it is difficult to reach this conclusion with the data presented in this paper, although other studies could be mentioned to support it. It would be important to discuss the results in Matamales et al., 2016, where authors performed a very similar DTR-induced ablation of ChAT neurons in the dorsal striatum of mice, and they too studied its effects on action control. In that study it was also found that there was no change in the establishment of initial instrumental contingencies, but the ChAT-depleted mice were unable to update these contingencies upon reversal. This paper is relevant and supports the role of ChAT neurons in contextual modulation of action.

6) On the other hand, the authors found that PV ablation dramatically impaired instrumental behaviours (Figure 7K), and also reported that these mice show virtually no sensitivity to stimuli. However, would the primary impairment in instrumental learning not preclude performance during the go/no go test? Can the conclusion "PV interneurons are important for stimulus-response associations" be formulated if animals show very limited instrumental capacity? This is perhaps worth discussing a bit.

7) Figure 8F is not referred to in the text.

---

## [Author Response]

Essential revisions:While all three reviewers appreciated the sheer amount and also the high quality of much of the data, they also felt that overall the paper came across as somewhat inconclusive or unfocused. On discussion, it was generally agreed that this was due to the lack of consistency in the behavioral approaches and the results. In contrast to the anatomical and ex vivo electrophysiological data, the behavioral experiments tended to suffer from major weaknesses (inconsistent designs, lack of opto controls, use of primarily excitatory methods, non-physiologic parameters) that made their interpretation difficult and/or there were inconsistencies across experiments that were difficult to resolve. These issues are evident across the reviews. While the authors may feel that these can all be resolved, the consensus of the reviewers was that the paper would be much better if it were refocused on the anatomical comparisons, with the behavioral results removed. This change, along with some adjustments to the comparisons (point #2, reviewer #3), were judged to be the essential revisions. We realize this is a large change, but if the authors are willing to do this, the reviewers felt a revision would be worthwhile.

We thank the reviewers for their detailed review of the manuscript and for their comments, which helped to improve the quality as well as refocus and streamline the manuscript. We removed all behavioral experiments from the manuscript, removed nearly all statistical comparisons between ChAT and PV tracing datasets instead refocusing on within group comparisons and extensively revised the manuscript to address all the reviewers’ concerns. We have removed three main figures and updated three main figures. Additionally, we have removed two supplementary figures, updated one supplemental figure and added a new supplemental figure. More specifically, the manuscript has been updated in the following:

1) We have removed all behavioral data including TRN terminal stimulation locomotion and accelerometer data (Figure 4), ICSS data (Figure 5), and ChAT and PV diphtheria-mediated ablation data (Figures 7 and 8), as well as three supplemental figures including cortical layer breakdown by cortical subregion (Supplementary Figure 4) and two diphtheria-mediated lesion behavior supplementals (Supplementary Figures 9 and 10).

2) All statistical comparisons between ChAT and PV including main Figures 2, 3, 4 and Supplementary Figure 4 are removed (except for Figure 2A, see the explanation below) and we have emphasized only the within-ChAT or PV comparisons throughout the results.

3) We have extensively revised the manuscript from the title to Discussion to integrate all of the changes mentioned above.

4) We have added images of parvalbumin and somatostatin staining of TRN to address the reviewer’s concern on the expression of parvalbumin and somatostatin neurons in TRN (Figure 3—figure supplement 1E, F).

5) We have added a new supplementary figure to address in vivo recording array placement (Figure 3—figure supplement 2).

6) We have added within group laminar cortical statistical comparisons (Results, subsection “Monosynaptic tracing reveals the inputs to striatal ChAT and PV interneurons”, last paragraph) and added some commentary on the similarities in the inputs of cortical laminar organization between interneuron and striatal D1-/D2- or patch-/matrix-projection neurons in associative and sensorimotor cortex (Discussion, subsection “Differential excitatory inputs to ChAT and PV striatal interneurons” last paragraph).

7) We have added a new paragraph to the discussion on the PPN excitatory inputs to ChAT interneurons comparing and contrasting against known PPN cholinergic inputs to striatum (Discussion, subsection “An excitatory pedunculopontine nucleus input to striatal ChAT interneurons” first paragraph).

We hope the reviewers now find that these revisions have addressed all the concerns and the manuscript is now suitable for publication in *eLife*.

Reviewer #1:[…] 1) I don't think that the functional data regarding the thalamic reticular nucleus inputs to PV neurons say very much. Here the authors stimulated these inputs in freely moving mice and found that locomotion was increased. However, there is no way to tell what this might mean. In the Discussion, this result is linked to a role for this input in modulation of attention, biasing action selection, and coordinating sensory attention and action. But really the result does not provide evidence supporting any of those ideas one way or the other.

Thank you for the comment. Our intention was simply to functionally validate the inhibitory TRN projection to striatal PV interneuron in freely moving animals. We found stimulation of TRN terminals in striatum facilitates locomotion. Following the reviewer’s suggestion, however, we have now removed that figure and all the TRN behavioral data from the manuscript.

2) The external segment of the globus pallidus inputs onto CINs are suggested to be involved in generating CIN pauses and possibly in reinforcement learning. But pause-like responses occur when these inputs are stimulated continuously for nearly a second, which is unlikely to be a physiologically relevant mechanism.

We thank the reviewer for the feedback on the stimulation parameters for GPe terminal stimulation. The 1-s stimulation duration was initially chosen because we found that it is efficient for supporting ICSS learning behavior. While we agree that 1-s constant terminal stimulation is not necessarily physiologically relevant, we would like to point out that inhibition of tonic firing in ChAT and PV occurs within the first tens of milliseconds following laser on. We have confirmed that this is the case for suppressing tonic basal firing in ChAT with fast latency in both ex vivo brain slice (Figure 4D, E and Figure 4—figure supplement 2H, I) and in in vivo recordings (Figure 4J). We have now removed the ICSS behavioral data based on reviewers’ suggestions and left only the electrophysiological data in Figure 4.

The authors also show that mice will learn to push a lever to stimulate these inputs and in a separate experiment to inhibit CINs. These results are interesting and suggestive. However, the latter effect is only about one tenth the magnitude of the former effect, suggesting that some other mechanism besides inhibiting CINs must be in play. I think the authors should comment on the disparity between the magnitude of these two results. What other mechanism (besides inhibition of CINs) might account for the highly robust self-stimulation of these inputs?

We thank the reviewer for the comment. We agree with the reviewer and have stated in the main text that direct ChAT inhibition only partially recapitulates the effects of GPe terminal stimulation in striatum in terms of ICSS magnitude. It is possible that GPe terminal stimulation inhibits many different striatal neurons including various interneurons and SPNs, and this inhibition works in concert with possible dopamine release evoked by ChAT interneuron rebound activity through nAChR on dopamine terminals (Cragg, 2006) for supporting ICSS behavior. In comparison, direct inhibition of ChAT interneurons will be restricted to a rather small number of ChAT cells due to the sparse distribution of ChAT interneurons in dorsal striatum (1-2%) and limited spread of light. However, we have decided to remove the ICSS dataset from the manuscript per the reviewers’ suggestion.

3) The behavioral effects of ablating CINs or PV neurons are suggestive and interesting, but again I think the conclusions reached are far from clearly supported. The PV-ablation effect on operant learning is very convincing. However, this does not seem consistent with what others have suggested in regard to PV neurons – e.g. a role in action selection. Can the authors clarify whether their data shed any light on why PV-ablation results in such an enormous learning deficit?

Thank you for the comment on the PV ablation data. We agree that striatal PV interneurons have been found to play a role in action selection (Gage et al., 2010). However, recent studies have expanded this role and suggested that striatal PV interneurons might be important in associative learning (Lee et al., 2017; Owen et al., 2018), consistent with what we found in our study. Nevertheless, we have removed the PV-ablation behavioral data from the manuscript according to the reviewers’ suggestion.

In the case of the CIN-ablation effect on the go/no go task, can the authors shed any additional light on why they interpret this as a deficit in context-dependent control of actions? Under this interpretation, why would the effect on baseline responding appear on day 1, before any learning of context has occurred in control mice? Wouldn't an alternative interpretation be that these rats have an unconditioned increase in food-elicited hyper-activity that they slowly learn to suppress?

We thank the reviewer for this comment. In our ChAT ablation data, rates of lever pressing are comparable between controls and lesion mice over seven days of continuous reinforcement (CRF) training. However, the difference of increased baseline press rate and magazine headentry rate in ChAT lesion versus controls appears on the first day of Go/No-Go training. We have now removed the ChAT and PV cell ablation behavioral data from the manuscript based on the reviewers’ suggestion.

Finally, two other questions/issues that I think the authors should address:a) They should clarify how many mice contributed to the rabies tracing experiment (sorry if I missed this), and where the placements were located. How did the authors confirm that the initial infusions of virus took place in identical locations in the PV-cre vs. Chat-cre mice?

We apologize for the confusion. We have moved the number of animals included in the rabies tracing earlier in the Results section and in the Materials and methods section. ChAT (N = 6) and PV (N = 5) mice were used in the rabies tracing experiments (Results, subsection “Monosynaptic tracing reveals the inputs to striatal ChAT and PV interneurons”, second paragraph; Materials and methods, subsection “Delta G-Rabies Tracing Viral Injections”). The coordinates and details for helper and rabies injections are found in the Materials and methods section under the header “Delta G Rabies Tracing Viral Injections”. The coordinates used were AP +0.5, ML -1.8, DV -2.25 to target dorsal central striatum. ChAT-Cre and PV-Cre mice were stereotaxic injected using the same coordinates and volumes of virus following a precise levelling of the skull in respect to bregma and lambda. Mice from both groups were injected within the same day. Rabies expression throughout dorsal striatum was confirmed in all brains prior to whole-brain cell counting and only included in the analysis if starter cells were contained within the dorsal striatum (see Figure 1—figure supplement 2 for example of starter cell confirmation and lack of cortical expression in negative controls).

Furthermore, are there any data in this experiment speaking to whether associative cortical inputs to CINs occur more in the dorsomedial locations in striatum or sensorimotor inputs to PV neurons occur more in dorsolateral locations? The authors show that overall the number of originator neurons in dorsomedial vs. dorsolateral are similar, but doesn't that proportion vary in each brain, and if so, wouldn't that shed some light on my question?

In all of our rabies-injected brains, the AAV helper injections covered a vast majority of the dorsal striatum that unfortunately limits our ability to disambiguate dorsal medial and dorsal lateral tracing to ChAT or PV interneurons. Our goal was to recruit as many starter cells as possible given that the interneuron types only make up roughly 1% of the total striatal neuron population. An additional complication for this quantification is the fact that striatal subdivisions are a gradient (ventral medial to dorsal lateral) defined more by the inputs then sharply demarcated territories with well-defined boundaries (Voorn et al., 2004). While there was variance in the number of starter cells, in our experiments the distribution of starters was similar across dorsal medial and dorsal lateral territories (Figure 1—figure supplement 2, determined with a simple bisecting line between dorsal medial and dorsal lateral striatum).

b) Can the authors comment on why they might have found that the pedunculopontine inputs to striatum are not cholinergic, in contrast to previously published data?

We thank the reviewer for this question. Our data does not preclude the existence of a cholinergic input from the PPN to striatum that has been previously reported in rats (Dautan et al., 2014). The lack of rabies labeled cholinergic neurons in this study using mice may potentially be due to a technical limitation with rabies tracing of neuromodulatory inputs because of their non-conventional synapses. For example, several striatal tracing studies using the same technique do not find significant rabies labeling of dopamine neurons in the substantia nigra pars compacta (SNc) (Smith et al., 2016; Wall et al., 2013), despite the SNc densely innervating the dorsal striatum. We assume that one reason could be that cholinergic projections from PPN may not label well with the rabies transsynaptic tracing system. Alternatively, the PPN cholinergic projection may not synapse directly on striatal ChAT interneurons. The selectivity of this system may give some hints as to why there are at least two sources of acetylcholine in striatum. We have added a paragraph in the Discussion to address these points (subsection “An excitatory pedunculopontine nucleus input to striatal ChAT interneurons”, first paragraph).

Reviewer #2:[…] The authors also use optogenetics to examine the function of three prominent inputs to PV or ChAT cells: a previously unknown GABAergic projection from the thalamic reticular nucleus, a projection from GPe, and a projection from the pedunculopontine nucleus. While I applaud the authors' intentions and recognize the huge effort that went into these experiments, I felt that these behavioral experiments were not well designed because there were some big inconsistencies in behavioral tasks, and the controls were sometimes not defined or poorly chosen. Additionally, in my reading the cell ablation study at the end seems to contradict some of the optogenetics results. For example, in Figure 4 they show that activating the GABAergic TRN projection to striatum-which preferentially targets PV cells-increases distance traveled. But in Figure 7 they show that genetically ablating PV cells doesn't affect distance traveled but it does affect time in center of arena (a measure of anxiety). Not only do these results seem contradictory, but the data the authors choose to plot in Figures 4 and 7 are inherently different.

We apologize for the confusion on the behavioral data. Our intention was to assign some functional behavioral relevance to the brain-wide inputs to striatal interneurons. We thank the reviewer for the pointing out the possible inconsistency on the role of PV interneurons and the locomotion. One possible reason underlying this difference is the different time course of inhibition vs. ablation, which may play an important role in determining the final behavioral outcome. TRN facilitation of locomotion (potentially via inhibition of PV interneurons) occurs on the order of seconds, while PV cell ablation evolves over the course of weeks that might allow circuit-level compensation to occur. In regards to the differences in plotting the data in Figures 4 and 7, we intended to use accelerometer which samples at 1kHz, much faster than video tracking, to show transient locomotion initiations. Nevertheless, we have responded to the reviewers’ suggestions by removing all behavioral data in the manuscript.

I was also confused why the authors used an ICSS task in Figure 5, but a food reward task in Figure 7. I also had a hard time finding information about number of animals or how the control experiments were designed.

Sorry for the confusion on the choice of behavioral tasks. Our intention in choosing an ICSS task in the previous Figure 5 was due to the fact that optogenetic GPe terminal stimulation was sufficient to initiate a pause-rebound firing pattern in ChAT interneurons. Given the literature on acetylcholine-mediated dopamine release from dopamine terminals in striatum (Sulzer et al., 2016), we hypothesized that GPe terminal stimulation may trigger dopamine release as well. However, we have responded to the reviewer’s concerns by removing all behavioral data in the manuscript.

There were also few if any GFP or YFP control experiments for the optogenetics part. Finally, the authors over-relied on optogenetic excitation of different inputs, which raises the concern about showing sufficiency not necessity.

While we did not include eGFP/eYFP controls in our locomotion studies, we agree with the reviewer that they are excellent controls for the effects of laser stimulation and the effect of light cue on behavior. However, we did use 5Hz optogenetic laser stimulation serving as a within animal control for laser stimulation in that particular experiment. We did not observe any significant changes in locomotion following 5Hz stimulation versus preceding baseline. Also, we have gone to great lengths to shield the light at the ferrule to optic fiber connection to avoid a visual laser light cue. We agree with the reviewer that in order to demonstrate the necessity besides sufficiency for different inputs, it would require optogenetic inhibition experiments, which in many cases might be harder to conduct due to the distributed input cells. Nevertheless, based on the reviewers’ suggestion, we have now removed all the behavioral data from the manuscript to keep it more focused.

I don't think the manuscript in its current form can be published without major changes to the behavioral experiments, or elimination of the behavioral experiments section altogether and publication of just the input tracing part.

Thank you for the suggestion. We have now removed all of the behavior from the manuscript and re-focused on the rabies-mediated monosynaptic tracing and functional electrophysiological dissection as you advised.

Reviewer #3:[…] I have however a few observations that would be worth clarifying/discussing before this manuscript reaches its final form:1) The finding that the same number of starter cells are present in the two Cre lines is puzzling (Figure 1—figure supplement 2E-G). Does this mean that the actual density is the same in both populations? This contrasts with the view that PV neurons are more frequent than ChAT neurons in the dorsal striatum. Perhaps it would be worth contrasting those numbers with those obtained in WT mice, just to rule out that Cre expression does not limit the proliferation of any of the two populations, as seems to be happening in certain lines (e.g. Harno et al. Cell Metabolism, 2013, 18 (1); 21-28). This is only a suggestion that might help reconciling with previous literature.

We thank the reviewer for the suggestion regarding the number of starter cells. Differences have been observed via stereological cell counting in the literature in the number of ChAT and PV interneurons observed in dorsal striatum. Yet, those numbers have varied from study to study and are also dependent on species (Oorschot et al. 2013). While the absolute number is up for debate the overall comparison between the number between ChAT and PV is more similar than different. Unbiased stereological cell counts of immunostained striatum suggest minimum estimates of 1-2% of ChAT interneurons versus 0.7% of PV interneurons (Luk and Sadikot, 2001; Rymar et al., 2004). Yet others have found estimates of 0.43% ChAT interneurons versus 0.59% PV interneurons in the rat dorsal striatum (Oorschot et al. 2013). The total counts we obtained of immunostained ChAT and PV interneurons in dorsal striatum was not significantly different (average number of immunopositive neurons per brain slice in striatum, N=7 per group, taken from 10 brain slices (40µm thick) spanning the injection site; ChAT 207.6 ± 12.63, PV 170.7 ± 47.46; two-tailed unpaired t-test, p = 0.8048. Given the similar counts of interneurons, the fact we find a comparable number of starter cells seems consistent.

2) In many instances in the paper, the percent of total inputs to PV and ChAT interneurons is statistically compared across the two systems side by side (e.g. Figure 1—figure supplement 2F-G; Figure 2B-F; Figure 3A-C; Figure S4A-L; Figure 4—figure supplement 1A). I am unsure about the validity of this comparison. To me, even if the number of starter neurons is similar and the area is matching, these two subpopulations form entirely different systems with very dissimilar physiological and morphological characteristics (e.g. Kawaguchi, JNeurosci, 1993,13(11); 4908-23). Features like dendritic arborisation, synaptic density and overall cellular volume may influence to a large extent the efficiency of trans-synaptic infection, therefore providing a very different pattern of labelling in circuits inputting onto PV and ChAT neurons. While this provides very valuable information on the relative proportion of afferent projections from diverse brain regions to each interneuronal system separately, I fail to see how comparing across the two systems is informative. Thus, claims such "ChAT neurons received significantly more inputs from the associative cortex than PV neurons" would be misleading. I think that all data should be presented (and statistical comparisons done) in each system separately-comparisons like "ChAT neurons received significantly more inputs from associative than sensorimotor cortices whereas PV neurons showed the opposite trend" would be appropriate. This would also reduce the amount of statistical comparisons, and would allow using within-subjects analyses which can be more powerful.

We thank the reviewer for the question regarding the validity of the anatomical comparison between percent of total inputs to ChAT versus PV interneurons. While we agree with the reviewer and acknowledge the differences in dendritic arborization, synaptic density and overall cellular volume between the two interneuron types, the side-by-side comparisons we conducted have been following a common practice in the field in which many researchers seemly still find normalized comparisons informative (Beier et al., 2015 Cell; Faget et al., 2016 Cell Reports; Kim et al., 2015 Neuron; Ogawa et al., 2014 Cell Reports; Smith et al., 2016; Wall et al., 2013; Watabe-Uchida et al., 2012 Neuron; Weissbourd et al., 2014 Neuron). Please refer to Figure 4G of Owaga et al., Figure 2D of Weissbourd et al. and Figure 3H of Faget et al.

However, following the reviewer’s suggestion, we have now removed all statistical comparisons between ChAT and PV interneurons throughout the main Figures 2 and 3, and completely removed the Supplementary Figure 4. In the manuscript we have now focused mainly on the within-group comparisons (Figure 2B-F and Figure 4—figure supplement 1). However, we have left the significance markers in Figure 2A for the readers in the field to make their own judgments without us making any major claims in the main text. Please let us know if it is ok or if you would like us to remove it too.

3) In the TRN study (subsection “Different thalamic projections to striatal ChAT vs. PV interneurons”, second paragraph), it is not immediately evident that parvalbumin and somatostatin staining inform about the anatomical boundary of the TRN, while this is a critical point in later viral manipulations (i.e. Figure 3A). It would be helpful to clarify this a bit more in the text, perhaps providing citations that describe the PV and SOM markers in TRN.

We thank the reviewer for calling this to our attention. TRN PV and SOM cells are seen throughout the AP axis of the TRN. The only unique topography is in the number of SOM-positive cells in the central tier of the middle sector of the TRN is lower than the number of PV-positive cells. Very few neurons display both PV and SOM markers (10-20%) in the somatosensory TRN (Clemente-Perez et al., 2017). We have added parvalbumin and somatostatin immunostaining of TRN to Figure 3—figure supplement 1E, F. Additionally, we have added a citation referencing the two main types of TRN neurons (PV and SOM) in the Results section of the manuscript (Clemente-Perez et al., 2017) (subsection “Thalamic projections to striatal ChAT and PV interneurons”, second paragraph).

4) While the full characterisation of the transgenic lines presented at the very start of the paper is appreciated, I find that some viral spread and/or multielectrode placement confirmation may be lacking in some of the in vivo experiments. For example the TRN and GPe multielectrode/optogenetics experiments (Figure 4, Figure 3—figure supplement 3, Figure 5, Figure 4—figure supplement 2) do not have placement verifications. Adding a diagram with the locations in each experimental animal could be helpful.

We thank the reviewer for pointing this out. We have added Figure 3—figure supplement 2 with representative in vivo recording sites for TRN to dorsal striatal recordings or GPe to striatal recordings to address this concern.

5) The toxicogenetics experiments seem very clean and are a very good option to expose the role of each interneuronal system in action control. In the case of ChAT ablation, the authors did not find any deficit in the initial acquisition of instrumental behaviours (Figure 7), and they found an increase of baseline lever-pressing rate which had no consequence on later stimulus-response learning. The authors argue that these data support that ChAT interneurons play a role in providing context to the modulation of action. However, I think that it is difficult to reach this conclusion with the data presented in this paper, although other studies could be mentioned to support it. It would be important to discuss the results in Matamales et al., 2016, where authors performed a very similar DTR-induced ablation of ChAT neurons in the dorsal striatum of mice, and they too studied its effects on action control. In that study it was also found that there was no change in the establishment of initial instrumental contingencies, but the ChAT-depleted mice were unable to update these contingencies upon reversal. This paper is relevant and supports the role of ChAT neurons in contextual modulation of action.

We thank the reviewer for the comment on our ChAT ablation behavior and the comparison to the Matamales et al. (2016) ChAT ablation behavior. While our diphtheria-mediated ChAT interneuron ablation mice are able to successfully learn CRF and the Go/NoGo task similar to controls, we noted a persistent elevated baseline press rate after transition from CRF to Go/NoGo training. This inability to properly modulate behavior following a rule change (each press leads to reinforcer versus one press within 10 seconds of “go” tone leads to reinforcer) may potentially signal an inability to update behavior given a change in context. We did perform reversal experiment where we switched the “go” and “no go” tones in the presence of CNO activation of a Gi-coupled DREADD to suppress ChAT interneuron activity. However, we did not observe any significant differences following reversal between control and experimental ChAT inhibition groups in learning. It is difficult to completely interpret a negative result, including whether ChAT interneurons were significantly inhibited with CNO or clozapine during the experiment, so we did not include these experiments in the manuscript. Nevertheless, we have now removed all behavioral data from the ChAT- and PV-ablation experiments based on the reviewers’ advice.

6) On the other hand, the authors found that PV ablation dramatically impaired instrumental behaviours (Figure 7K), and also reported that these mice show virtually no sensitivity to stimuli. However, would the primary impairment in instrumental learning not preclude performance during the go/no go test? Can the conclusion "PV interneurons are important for stimulus-response associations" be formulated if animals show very limited instrumental capacity? This is perhaps worth discussing a bit.

We thank the reviewer for the comment on the PV ablation data set. We concede that the PV ablation mediated disruption in CRF learning would preclude any subsequent stimulus response learning. However, due to variability in potentially the degree of lesion we had a subpopulation of PV ablation mice that were able to learn CRF, but were unable to learn the go/no go discrimination task. This role of PV interneurons in early associative learning is reinforced by two recent publications (Lee et al., 2017; Owen et al., 2018). However, we have now removed the diphtheria toxin-mediated ablation of PV interneurons experiments per the reviewers’ suggestion.

7) Figure 8F is not referred to in the text.

Thanks for noting. We have now removed Figure 8 and all the behavioral experiments.